# The structural pathology for hypophosphatasia caused by malfunctional tissue non-specific alkaline phosphatase

Yating Yu[1,2,5], Kewei Rong[1,5], Deqiang Yao[3,5], Qing Zhang [1,2], Xiankun Cao[1], Bing Rao[1,2], Ying Xia[2], Yi Lu[2], Yafeng Shen[2], Ying Yao[4], Hongtao Xu [4], Peixiang Ma [1] ✉, Yu Cao [1,2] ✉ & An Qin [1] ✉

Hypophosphatasia (HPP) is a metabolic bone disease that manifests as developmental abnormalities in bone and dental tissues. HPP patients exhibit hypomineralization and osteopenia due to the deficiency or malfunction of tissue non-specific alkaline phosphatase (TNAP), which catalyzes the hydrolysis of phosphate-containing molecules outside the cells, promoting the deposition of hydroxyapatite in the extracellular matrix. Despite the identification of hundreds of pathogenic TNAP mutations, the detailed molecular pathology of HPP remains unclear. Here, to address this issue, we determine the crystal structures of human TNAP at near-atomic resolution and map the major pathogenic mutations onto the structure. Our study reveals an unexpected octameric architecture for TNAP, which is generated by the tetramerization of dimeric TNAPs, potentially stabilizing the TNAPs in the extracellular environments. Moreover, we use cryo-electron microscopy to demonstrate that the TNAP agonist antibody (JTALP001) forms a stable complex with TNAP by binding to the octameric interface. The administration of JTALP001 enhances osteoblast mineralization and promoted recombinant TNAP-rescued mineralization in TNAP knockout osteoblasts. Our findings elucidate the structural pathology of HPP and highlight the therapeutic potential of the TNAP agonist antibody for osteoblast-associated bone disorders.

The establishment of an inorganic extracellular matrix is critical in the biomineralization progress. The formation of hydroxyapatite crystal within the matrix vesicles of osteoblasts and odontoblasts initiates the mineralization process and leads to the incorporation of hydroxyapatite with the collagen into the extracellular matrix on the surface of osteoblasts and odontoblasts[1–3]. Several proteins enriched in the matrix vesicles supply sufficient phosphate for the crystallization of hydroxyapatite, and among them, the TNAP catalyzes the hydrolysis reaction using the pyrophosphate and organic phosphate esters as the substrates, such as pyrophosphate (PPi), pyridoxal phosphate (PLP) and phosphoethanolamine (PEA)[4–6]. In addition to the phosphate supplies, TNAP could modulate bone growth by controlling the PPi level, since the PPi can bind to the surface of the hydroxyapatite crystal to form a "coating" layer, which prevents both further calcifications in undesired environments such as plasma and the dissolution of the mineral materials from bone and teeth[7]. During the process of bone

[1]Department of Orthopedics, Shanghai Key Laboratory of Orthopedics Implant, the Ninth People's Hospital, Shanghai Jiao Tong University School of Medicine, Shanghai 200011, China. [2]Institute of Precision Medicine, the Ninth People's Hospital, Shanghai Jiao Tong University School of Medicine, 115 Jinzun Road, Shanghai 200125, China. [3]State Key Laboratory of Oncogenes and Related Genes, Ren Ji Hospital, Shanghai Jiao Tong University School of Medicine, Shanghai 200127, China. [4]Shanghai Institute for Advanced Immunochemical Studies, ShanghaiTech University, Shanghai 201210, China. [5]These authors contributed equally: Yating Yu, Kewei Rong, Deqiang Yao. ✉e-mail: mapx@shsmu.edu.cn; yu.cao@shsmu.edu.cn; dr_qinan@163.com

formation, TNAP could act to remove this pyrophosphate coating to keep the growth of the bone extracellular matrix.

The disruption of TNAP function results in insufficient bone mineralization, e.g., hypophosphatasia (HPP)[8]. Hypophosphatasia is a heterogeneous disease characterized by a low level of skeletal mineralization. It has a broad range of severity, varying from life-threatening prenatal form, severe infantile form to odontohypophosphatasia form with premature loss of deciduous teeth. This disorder is genetically associated with inactivating mutations on alkaline phosphatase, tissue-nonspecific isozyme (ALPL) gene that causes deficient /dysfunctional enzymatic activities of TNAP. More than 400 pathological mutant alleles have been established, whilst their genotype/phenotype correlations are poorly understood. For perinatal and infantile forms of hypophosphatasia, the administration of human recombinant TNAP, Asfotase alfa, showed well tolerance and promising survival[9,10].

Alkaline phosphatases (AP) could be extensively found in microbes, animals, and plants. Four homologous AP genes exist in the human genome, ALPI, ALPL (TNAP), ALPP (PLAP), and ALPG, which encode the alkaline phosphatases of intestinal, tissue-nonspecific, placental, and germ cell isotype, respectively[11]. Precedent structural studies on Escherichia coli AP (ecPhoA) and human PLAP showed that AP forms a homodimer where each protomer could function as independent phosphatase using divalent metal ions as cofactors, such as $Zn^{2+}$ and $Mg^{2+}$, as well as a $Ca^{2+}$ ion binding site located away from the active pocket with unclear function[12–15]. Due to the established role of the mutations on TNAPs in the pathology of HPP, its structure information has been extensively pursued, but only homology modeling based on the structures of PLAP was available for the analysis of molecular pathology[16–18]. The molecular mode pyridoxal 5′-ling showed important features of TNAP such as a dimeric architecture and an active site with more polar residues in comparison with PLAP and also revealed the pathological effects of the mutations on the residues located in the critical regions, i.e., dimeric interface, active site and calcium-binding pocket[19–21]. However, with the current model, a series of questions remain to be addressed. Although the homology modeling could explain the pathogenicity for many HPP-related mutations, there are still quite a few mutations on the residues located away from the critical functional region of TNAP that have their molecular pathology beyond our current understanding. In addition to PPi, TNAP could catalyze the dephosphorylation reaction on various physiological molecules such as PLP, phosphocreatine (PC)[22], ATP, etc., and thus an accurate structural model is required to elucidate the substrate specificity and recognition of TNAP.

In this study, we solve the crystal structure of TNAP at near-atomic resolution and reveal an unexpected octamerization of TNAP with apparent biological function. Interestingly, the TNAP agonist antibody (JTALP001) binds to TNAP by the contacts on the octameric interface from the cryo-EM studies. Functionally, the administration of JTALP001 enhances osteoblast mineralization and promotes the recombinant TNAP-rescued mineralization in *Alpl* knockout osteoblasts. The structures reveal a molecular pathology of the HPP caused by the TNAP high-order oligomerization, providing a therapeutic potential for osteoblast-associated bone disorders.

## Results

### Crystallization and structure solving
Human TNAP (hTNAP) can be overexpressed and purified from *E. coli* and eukaryotic cells. However, the hTNAP protein prepared from *E. coli*-based overexpression system failed to show phosphatase activities in a biochemical assay. In contrast, insect-baculovirus and HEK293-based overexpression systems could generate hTNAP proteins with significant phosphatase activities at comparable levels (Fig. 1a–c and Supplementary Fig. 1a). Expression constructs were designed to express the amino acids 18–500 of hTNAP, the mature form of TNAP, with the GPI-anchor amidated residue Ser 501 removed to ensure an

efficient secretory expression. In a biochemical assay based on the colorimetric products from the enzymatic hydrolysis of p-Nitrophenyl Phosphate (pNPP), both the media from the hTNAP-overexpressing cell culture and the purified hTNAP therefrom showed phosphatase activities which are pH-dependent and could be inhibited by levamisole, a reported inhibitor against alkaline phosphatase (Fig. 1a, b)[23,24]. Further analysis using size-exclusion chromatography, chemical cross-linking assay, and native gel electrophoresis indicated the hTNAP could form oligomers with various degrees of polymerization, where dimeric hTNAP dominated in solution as estimated by the static light scattering analysis, with a significant proportion forming higher oligomers (Supplementary Figs. 1b, d and 12). Crystallization trials showed the crystals of purified hTNAP could grow in both acidic and basic pH conditions diffracting to about 5–10 Å resolution. After extensive optimization of crystallization conditions, the crystals of hTNAP growing in acidic conditions diffracted to about 2.9 Å and in basic conditions diffracted to about 3.2 Å, allowing the structure determination of hTNAP by molecular replacement using the molecular model of placental type alkaline phosphatase (PDB ID 1EW2).

### The dimeric structure of human TNAP
In previous crystal structures, alkaline phosphatases (APs) were found to form stable dimers[13,14]. For example, the crystal structures of ecPhoA and human placental AP (hPLAP) showed AP dimers with two zinc ions and one magnesium ion in ecPhoA[13,25], while hPLAP contains four metal ions, including three at the active site (two $Zn^{2+}$ and one $Mg^{2+}$) and one in a distal pocket, which was likely a calcium ion[14,17]. The human TNAP dimer in the crystal structure shows high structural similarity to ecPhoA and hPLAP in a structural superposition with the overall RMSDs of 1.625 Å between hTNAP and hPLAP, and 7.551 Å between hTNAP and ecPhoA (Fig. 1d and Supplementary Fig. 2). In each hTNAP protomer, the major body is made up of nine β-strands (strands a-i) sandwiched by two clusters of helices, hereinafter cluster-CA and -MZ. As shown in Fig.1d and Supplementary Fig. 3, the cluster-CA holds the $Ca^{2+}$ binding site with the helices 3, 4, 15–18, and 20, and the cluster-MZ holds the $Mg^{2+}$ / $Zn^{2+}$ binding sites with the helices 5–14, 22, and 23 (Supplementary Fig. 3a, b). In addition to the "helices-strands-helices" sandwich, the N-terminal of hTNAP formed by helices 1 and 2 extends into the neighboring protomer to stabilize the dimerization, and a crown-like domain is formed by helix 21, strand Cr1 and Cr2, and several loops at the other side of the dimeric interface (Supplementary Fig. 3a). The hTNAP dimerizes with a large dimeric interface which is comprised of the N-terminal helices, crown-like domain, helices 3, 4, 19, 22, and 23, as well as the strands b, h, and i (Fig. 1e and Supplementary Fig. 3c, d). The hTNAP dimer is maintained by the static electric interactions between R71-D458, as well as the hydrogen bonds and hydrophobic interactions among residues such as T68, T85, K99, T103, Q106, H341, Y388, T389, R391, D404, N417, S445, H454, V382, L414, V447, L449, and R450 (Fig. 1f). The role of dimerization in TNAP function is highlighted by the identification of pathologically related mutation sites among the interactive residues (Fig. 1f). For example, the missense mutations on R71, V382, R391, L414, N417, S445, and R450 was reported in HPP patients with dominant negative effect and symptom from moderate to severe[21,26–28] (Fig. 1f and Supplementary Data 1), implying the importance of dimerization in TNAP function (Fig. 1c).

### The catalytic and regulatory sites in human TNAP
The metal ions binding pockets were identified in the crystal structure of hTNAP, which along with the catalytic sites, are comprised of residues conserved across APs of different subtypes and species (Fig. 2a and Supplementary Fig. 2a). Both calcium binding site and triple metal binding sites were surrounded with a highly negatively-charged surface (Fig. 2b). The calcium-binding site includes the acidic residues E235, E291, and D306 forming

strong static electric interactions with the Ca$^{2+}$ ion, as well as R223, Y236, and T305 stabilizing the local conformation of the binding pocket (Fig. 2c). Four pathologically related mutation sites from patients with moderate to severe HPP were reported for charged residue R223, E235, E291, and D306[28–31] (Supplementary Data 1). In the functional assay, the HEK293T cell expressing the disease-associated mutants, D306V, E235G, and E291K, showed

significantly lower AP activities compared with wild-type hTNAP (Fig. 2e). In addition, the AP activity of hTNAP decreased upon adding EGTA into the reaction system, which could be rescued by supplementing calcium ions (Fig. 1a), suggesting a reversible regulation on the enzymatic activity by the calcium-binding.

As shown in Fig. 2d, the catalytic center of hTNAP is comprised of the catalytic residue S110 and three metal binding sites, one for Mg$^{2+}$

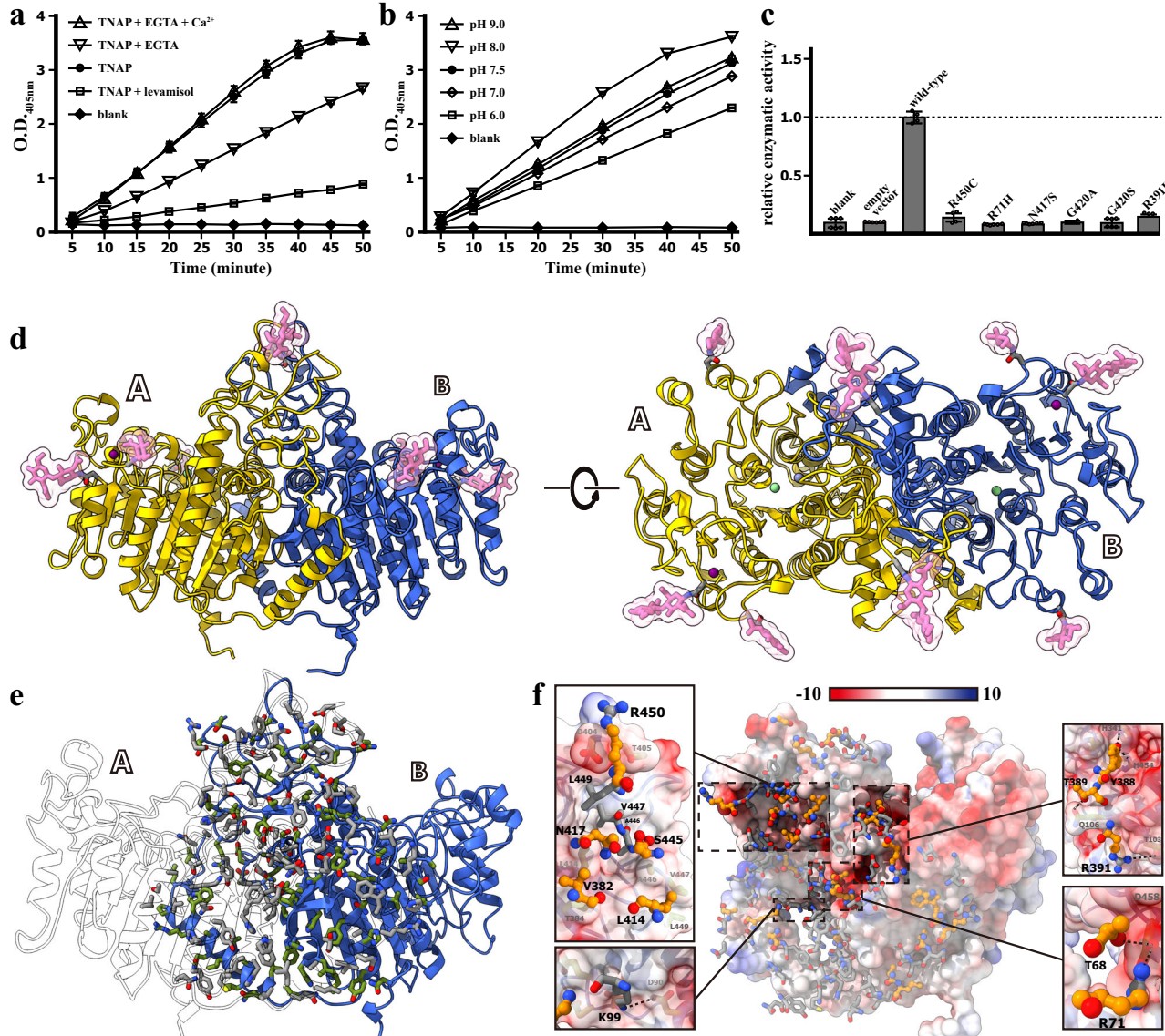

**Fig. 1 | hTNAP$^{18–500}$ functions as dimeric, pH-dependent phosphatase. a** The phosphatase activity of the purified hTNAP$^{18–500}$ protein. The TNAP activities were inhibited by either levamisole (0.9 mM) or EGTA (0.0182 mM) treatment, the latter effect was rescued by the re-supplement of calcium (0.9 mM). optical density (OD). $n$ = 3 biologically independent samples. **b** The hTNAP activities showed pH-dependence, which increased along with the pH of the reaction solutions. optical density (OD). $n$ = 3 biologically independent samples. **c** The disruptions of the hTNAP$^{18–500}$ activity by the mutations at the dimeric interface. The dimeric mutants of hTNAP$^{18–500}$ proteins (TNAP with mutations at the dimeric interface) were over-expressed in HEK293T and the phosphatase activities in cell media were measured and calibrated with the expression level as determined by the Western blotting (see Supplementary Fig. 8). $n$ = 6 biologically independent samples in blank, empty vector, R450C, R71H, N417S, G420S group. $n$ = 4 biologically independent samples in wild-type, G420A group. $n$ = 3 biologically independent samples in R391H group. **d** The crystal structure of hTNAP$^{18–500}$ dimer viewed from "side" (left) and "top" (right). Two TNAP protomers in the dimer were shown as cartoon model and

colored in yellow (A) and blue (B). The sugar moieties of N-glycosylation were shown as stick and surface model and colored in pink, with the associated asparagine residues shown as stick model. The metal ions found in TNAP were shown as balls and colored in purple (Ca$^{2+}$), green (Mg$^{2+}$) and gray (Zn$^{2+}$). **e** The dimeric interface of hTNAP$^{18–500}$. The TNAP dimer was shown as cartoon model with the protomer B colored in blue and A in transparency. The residues involved in dimerization were shown as stick model and color by elements (protomer A: gray-blue-red; protomer B: green-blue-red). **f** The interactions maintaining hTNAP$^{18–500}$ dimer. The protomer B was shown as calculated solvent-accessible electrostatic surface-potential maps and the interacting residues of the protomer A were shown as stick model and colored by elements, except that the residues reported with HPP related mutations were shown as stick-ball model and colored in orange-blue-red. Insets at side: the enlarged views showing the interactive sub-regions. All data in this figure are represented as mean ± SD. All experiments were repeated three times independently with similar results. Source data are provided as a Source Data file.

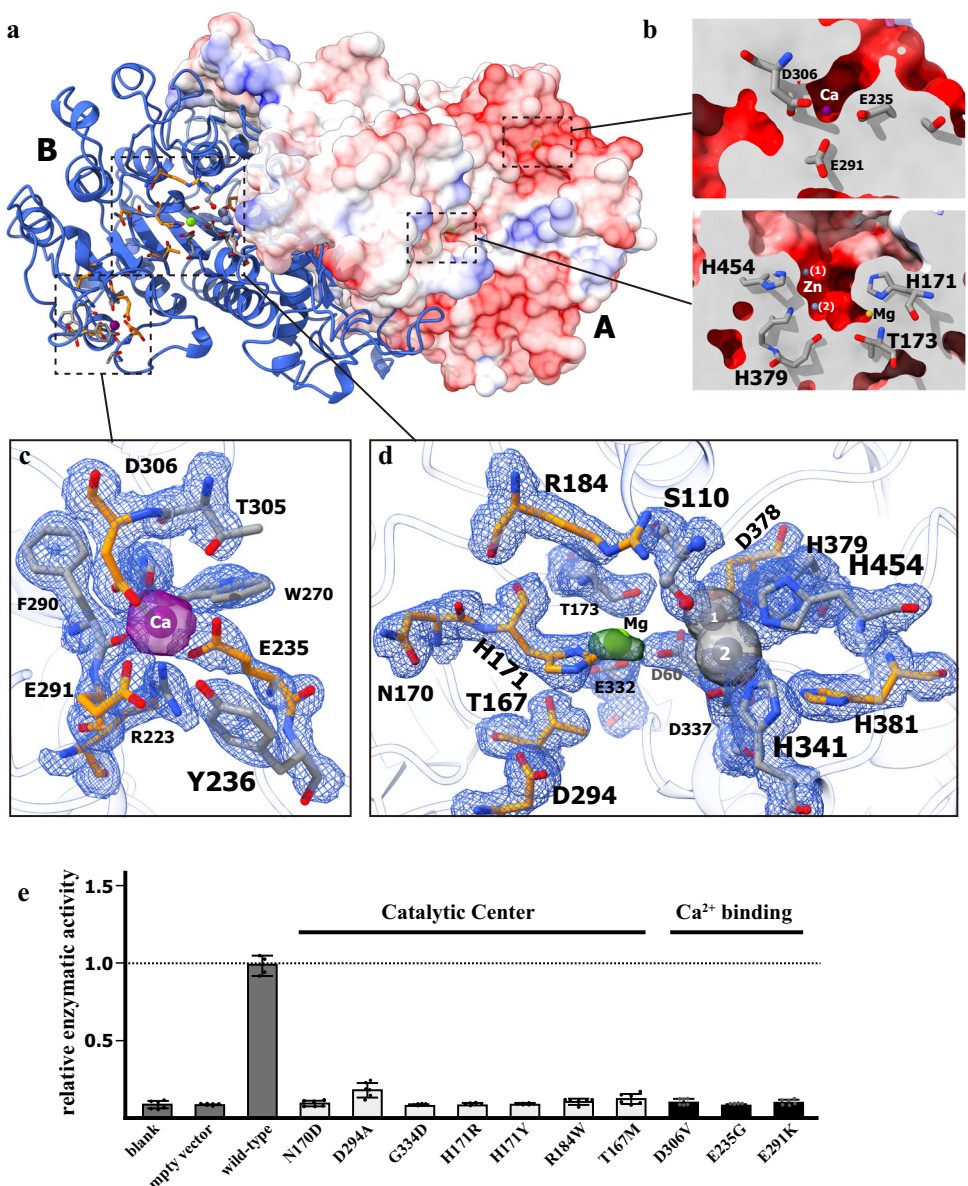

**Fig. 2 | The catalytic center and metal ion binding sites of hTNAP[18–500]. a** The "top" view of the dimeric hTNAP[18–500] showed the metal ions binding sites and the catalytic center. The protomer A was shown as electrostatic surface-potential maps transparent to show the Ca$^{2+}$-binding site and the Zn$^{2+}$/Mg$^{2+}$-binding site (indicated in dotted frames). The protomer B was shown as cartoon model, with the ions as balls and colored in purple (Ca$^{2+}$), green (Mg$^{2+}$), and gray (Zn$^{2+}$). The ion-interacting residues were shown as stick model and colored by elements. The residues involved in HPP were colored in orange-blue-red, and others in gray-blue-red. **b** The surface model of protomer B from Fig. 2a was sliced and zoomed in to show the ion binding pocket indicated in dotted frames. **c** The Ca$^{2+}$-binding site. The Ca$^{2+}$ ion was shown as purple ball within the omit Fo-Fc electron density map as purple solid surface (3.0 σ level). The interacting residues were shown as stick model colored in orange-blue-red (for HPP-related) and gray-blue-red within the omit Fo-Fc electron density map as blue mesh (1.5 σ level). **d** The reaction center and Mg$^{2+}$/Zn$^{2+}$-binding sites. The Mg$^{2+}$ ion was shown as green ball within the omit Fo-Fc electron density map as

green solid surface (1.5 σ level), and the two Zn$^{2+}$ ions (labeled as 1 and 2) as gray balls within the omit Fo-Fc electron density maps as gray surface (2.0 σ level). The interacting residues were shown as stick model colored in orange-blue-red (for HPP-related) and gray-blue-red within the omit Fo-Fc electron density map as blue mesh (1.5 σ level). The catalytic residue S110 was shown as stick-ball model. **e** The mutations in catalytic center and the Ca$^{2+}$ binding site exhibited no TNAP activity. The hTNAP[18–500] protein mutants were overexpressed in HEK293T and the phosphatase activities in the cell media were measured and calibrated with the expression level as determined by the Western blotting. $n = 6$ biologically independent samples in blank, empty vector, N170D, D294A, G334D, H171R, H171Y, R184W, T167M, D306V, E235G, E291K group. $n = 4$ biologically independent samples in wild-type group. All data in this figure are represented as mean ± SD. All experiments were repeated three times independently with similar results. Source data are provided as a Source Data file.

and two for Zn$^{2+}$ (1 and 2). The three ions bound in hTNAP are close to each other and surrounded majorly by the acidic residues D60, E332, D337, and D378, as well as five histidines H171, 341, 379, 381, and 454 (Fig. 2d). The interactive residues in the catalytic center form an electrostatic interactive network with the Mg$^{2+}$ and Zn$^{2+}$ ions, where D60 played a role in bridging residues between the Mg$^{2+}$ and Zn$^{2+}$ (2) and the two Zn$^{2+}$ ions form a positively-charged cage with a distal

residue R184 for the substrate molecules to stabilize its phosphate group and initialize the cleavage of the ester bond[13,32]. The conserved catalytic residue S110 was solved as the unphosphorylated form in all the protomers, which serve as a nucleophile in attacking the phosphorous (Fig. 2d). Among the residues coordinating with the metal ions and substrates, T167, H171, R184, D378, and H381 were shown associated with HPP and patients carrying mutations on those

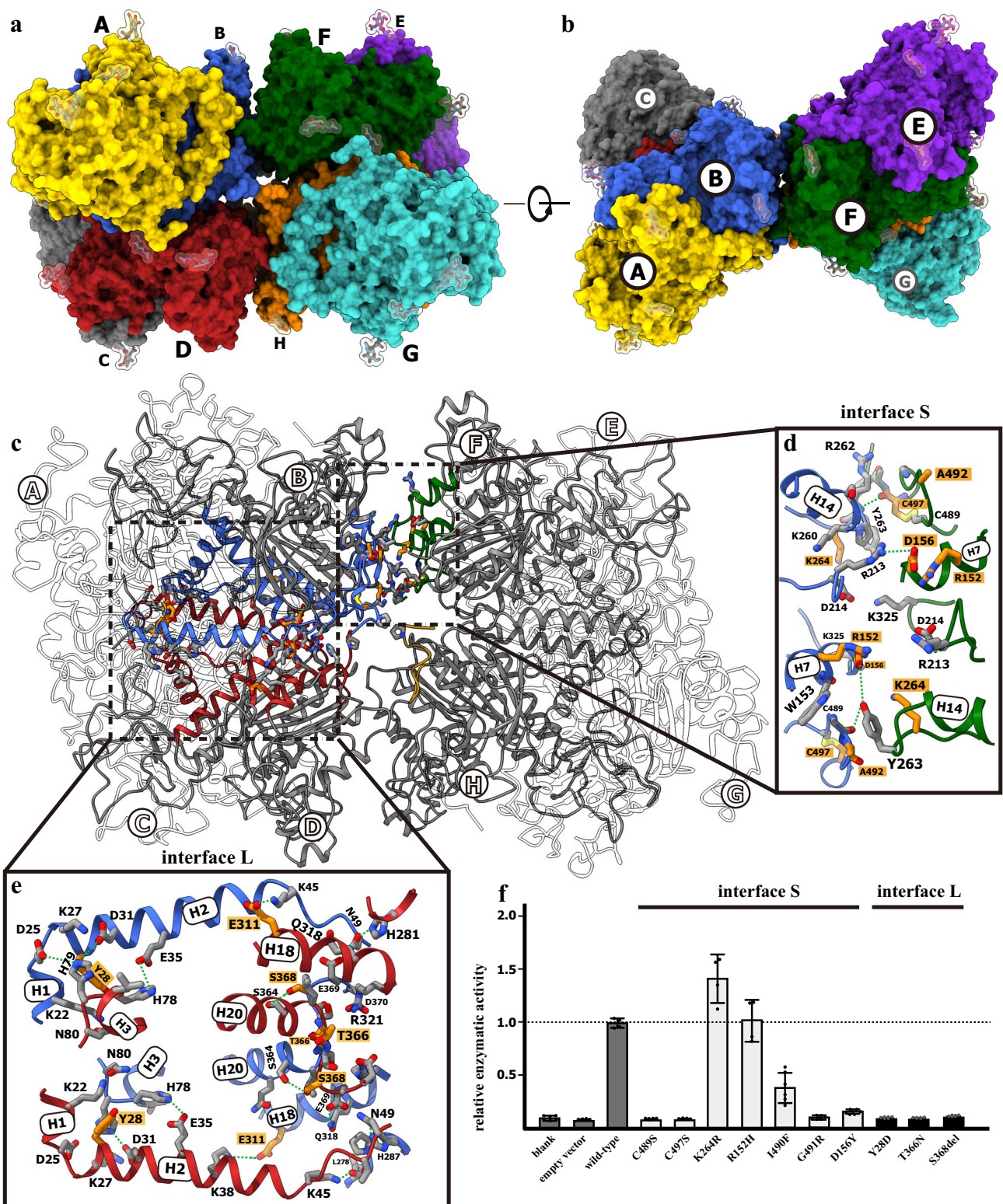

residues could develop severe bone symptoms[26,28–30,33,34] (Fig. 2e and Supplementary Data 1).

## The high-order architecture of human TNAP

In both structures determined with hTNAP crystals grown in acidic and basic pH conditions, four TNAP dimers, i.e., protomers A-B, C-D, E-F, and G-H, form an X-shape homo-octamer stabilized with the molecular contacts among protomers B, F, D, and H (Fig. 3a, b and Supplementary Figs. 4, 5). The oligomerization of alkaline phosphatases (APs) has

been observed since the 1980s[35–37], and the tetrameric or higher oligomeric TNAP was found in both organ-purified samples and recombinantly expressed TNAP[37–39]. The octameric structure observed in this crystal structure may represent one of the natural forms of TNAP. The TNAP octamer is maintained with four major interfaces between dimeric TNAPs. Two pairs of dimers, A-B/C-D and E-F/G-H, first form two tetramers with a large interface (interface L) comprising residues from helices 1–3, 18, and 20 via a series of static electric interactions and hydrogen bonds among H79 and D25/D31, E35, and H78, K38/K45

**Fig. 3 | The higher oligomeric hTNAP$^{18-500}$ structure. a, b** The hTNAP$^{18-500}$ octamer shown as surface model and viewed from two angles, respectively. The hTNAPs were colored by protomers and the glycosylation moieties were shown as stick and surface model. **c** The octameric interfaces of hTNAP$^{18-500}$. The secondary structural domains involved in octamerization were colored in blue, red, green, and gold for protomer B, D, F, and H, respectively. **d** The enlarged view showed the small interface (interface S) between protomer B and F. The residues mediating the intermolecular interactions were shown as stick model and colored by elements (orange-blue-red for HPP-related and gray-blue-red for others). **e** The enlarged view showing the large interface (interface L) between protomer B and D. The residues mediating the intermolecular interactions were shown as stick model and colored by elements (orange-blue-red for HPP-related and gray-blue-red for others). **f** The mutations in octameric interfaces showed varying alkaline phosphatase activities. The mutants of hTNAP$^{18-500}$ proteins were overexpressed in HEK293T and the phosphatase activities in the cell media were measured and calibrated with the expression level as determined by the Western blotting (see Supplementary Fig. 14). $n = 6$ biologically independent samples in blank, empty vector, C489S, C497S, I490F, G491R, D156Y, Y28D, T366N, S368del group. $n = 4$ biologically independent samples in wild-type, K264R, R152H group. All data in this figure are represented as mean ± SD. All experiments were repeated three times independently with similar results. Source data are provided as a Source Data file.

and E311, S364-S368, and T366-T366 from the contacting protomers (Fig. 3c, e). The tetramers A/B/C/D and E/F/G/H then dimerize to form the octamer with two small interfaces (interface S) comprising H7, H14, and some loops (Fig. 3c, d). Interestingly, multiple HPP-associated mutation sites were mapped in interface L and S, including D156, K264, A492, and C497 from interface S and Y28, E311, T366, and S368 from interface L (Fig. 3d, e). The further functional assay showed that the pathological mutants TNAPs carrying Y28D, D156Y, T366N, and C497S[19,20,40] could abolish or significantly reduce the enzymatic activities of TNAP, suggesting the importance of the octamerization in maintaining the stability of TNAP (Fig. 3f). Additionally, mutations on the oligomeric interface primarily affected the expression level of TNAP when overexpressed in HEK293T cells, except for the TNAP$^{K264R}$ mutant showing an expression level and AP activity comparable with that of wild-type TNAP.

## The TNAP agonist antibody binds to the octameric interface

The loss of TNAP activities by the mutations in the octamer interface might result from the structural instability upon the exposure of interfaces L and S. To study the potential function of octamerization, we developed antibodies specifically binding to hTNAPs. The TNAP agonist antibody (JTALP001) showed high affinity to hTNAP in ELISA assay, and the scFv fragment could form a stable complex with hTNAP in gel filtration (Supplementary Fig. 1c–e). In the cryo-electron microscopic study, the structure of the hTNAP-scFv complex was determined at a resolution of 2.96 Å, allowing a precise epitope mapping. In EM structure, the hTNAP maintains its dimeric form and binds with scFv at 1:1 stoichiometry, i.e., one TNAP dimer in complex with two scFv fragments (Fig. 4a). The binding of scFv on TNAP was primarily mediated by the interactions among scFv E132-TNAP K264, scFv S193-TNAP R213, scFv Y255-TNAP S258/R262, and scFv Y254-TNAP Y263 (Fig. 4b, c). The structural superposition between octameric TNAP and TNAP-scFv structures showed that the scFv epitope overlaps with interface S (Supplementary Fig. 6), resulting in the predominant form of dimer in EM analysis. Interestingly, the supplement of scFv in TNAP reaction solution significantly enhanced the phosphatase activity in the enzymatic assays on wild-type and mutants at the octameric surfaces. However, the scFv failed to activate TNAP mutants at the dimeric interface (R450C) or active site (T167M) (Fig. 4d, e), indicating the rescuing mechanism of JTALP001 depends exclusively on the protection of the oligomerization interface exposed.

## The dysfunctional TNAP mutants fail to promote osteoblast mineralization

To investigate the impact of different tissue non-specific alkaline phosphatase (TNAP) mutants on osteoblast differentiation and mineralization, we produced and purified recombinant TNAPs with mutations at the active site and dimeric interface. While the active site mutant TNAP$^{T167M}$ was purified as a stable dimer, as shown in SEC analysis (Supplementary Fig. 13), stabilizing TNAP in its monomeric form is challenging due to the high hydrophobicity of the dimeric interface, which makes it difficult to stabilize the protein in its monomeric form. Mutations at the dimeric interface often result in highly

unstable AP mutants expressed at very low levels[41-43] and difficult to purify. However, TNAP $^{R450C}$ mutation, which is associated with hypophosphatasia[26,28], was less deleterious and thus could be purified as a dimer (Supplementary Fig. 13). To assess the impact of these mutants on osteoblast differentiation and mineralization, wild-type TNAP, dimeric interface mutant TNAP$^{R450C}$, active site mutant TNAP$^{T167M}$, interface L mutant TNAP$^{S368A}$, and interface S mutants TNAP$^{R152H}$, TNAP$^{K264R}$ were subjected to the osteoblast differentiation and mineralization assay. As shown in the assays with either Alizarin Red staining or Von Kossa staining (Fig. 5), recombinant wild-type TNAP, TNAP$^{S368A}$, TNAP$^{R152H}$, and TNAP$^{K264R}$ proteins at 600 ng/ml enhanced the mineralization of the MC3T3-E1 cell line when compared with the control group, while neither TNAP$^{R450C}$ nor TNAP$^{T167M}$ could significantly induce MC3T3-E1 mineralization. In addition, recombinant wild-type TNAP, as well as TNAP$^{S368A}$, TNAP$^{R152H}$, and TNAP$^{K264R}$ protein, also promoted human bone marrow-derived mesenchymal stem cells differentiation to osteoblasts and subsequent mineralization, while TNAP$^{R450C}$ or TNAP$^{T167M}$ exhibited no effect on osteoblast mineralization (Fig. 5).

## The TNAP agonist antibody JTALP001 promotes osteoblast mineralization

To probe the agonist effect of JTALP001, we used $Alpl^{-/-}$ mice-derived pre-osteoblasts to further exclude the effect of endogenous TNAP. The knockout of $Alpl$ was confirmed by genotyping and the western blotting analysis of TNAP protein (Supplementary Fig. 7a, b). The alkaline phosphatase staining assay confirmed attenuated osteogenesis in calvaria-derived pre-osteoblasts from KO mice (Supplementary Fig. 7c, d). The administration of recombinant wild-type TNAP protein rescued osteoblast mineralization. In agreement with our previous alkaline phosphatase activity assay, wild-type TNAP, TNAP$^{S368A}$, TNAP$^{R152H}$, and TNAP$^{K264R}$ protein rescued mineralization in $Alpl^{-/-}$ mice-derived pre-osteoblasts but the administration of TNAP$^{R450C}$ or TNAP$^{T167M}$ recombinant protein failed to restore osteoblast mineralization (Fig. 5a, e, h and Supplementary Fig. 9).

Based on the findings that the JTALP001 antibody could enhance the phosphatase activity, we further examined the effect of this TNAP agonist antibody on osteoblast mineralization in vitro. The fluorescence-activated cell sorting (FACS) analysis was performed to evaluate the interaction between antibody and hTNAP proteins expressed on the cell surface. The FACS results confirmed the recruitment of JTALP001 to the cells overexpressing hTNAP (Supplementary Fig. 11). As shown in Fig. 6a–c, Alizarin Red staining and Von Kossa staining showed that the JTALP001 promoted the mineralization of human mesenchymal stem cells. In agreement with this, the phosphatase activity assay on whole cell lysates also confirmed that JTALP001 enhanced AP activity (Fig. 6d). Furthermore, the effect of JTALP001 was also validated in pre-osteoblasts from $Alpl^{-/-}$ mice. Osteogenesis and mineralization were enhanced by the administration of wild-type TNAP, TNAP$^{S368A}$, TNAP$^{R152H}$, and TNAP$^{K264R}$ protein. The administration of JTALP001 further boosted osteoblast mineralization compared to the control group using the TNAP and Fc fragment combination. Consistent with previous findings that TNAP$^{R450C}$ or

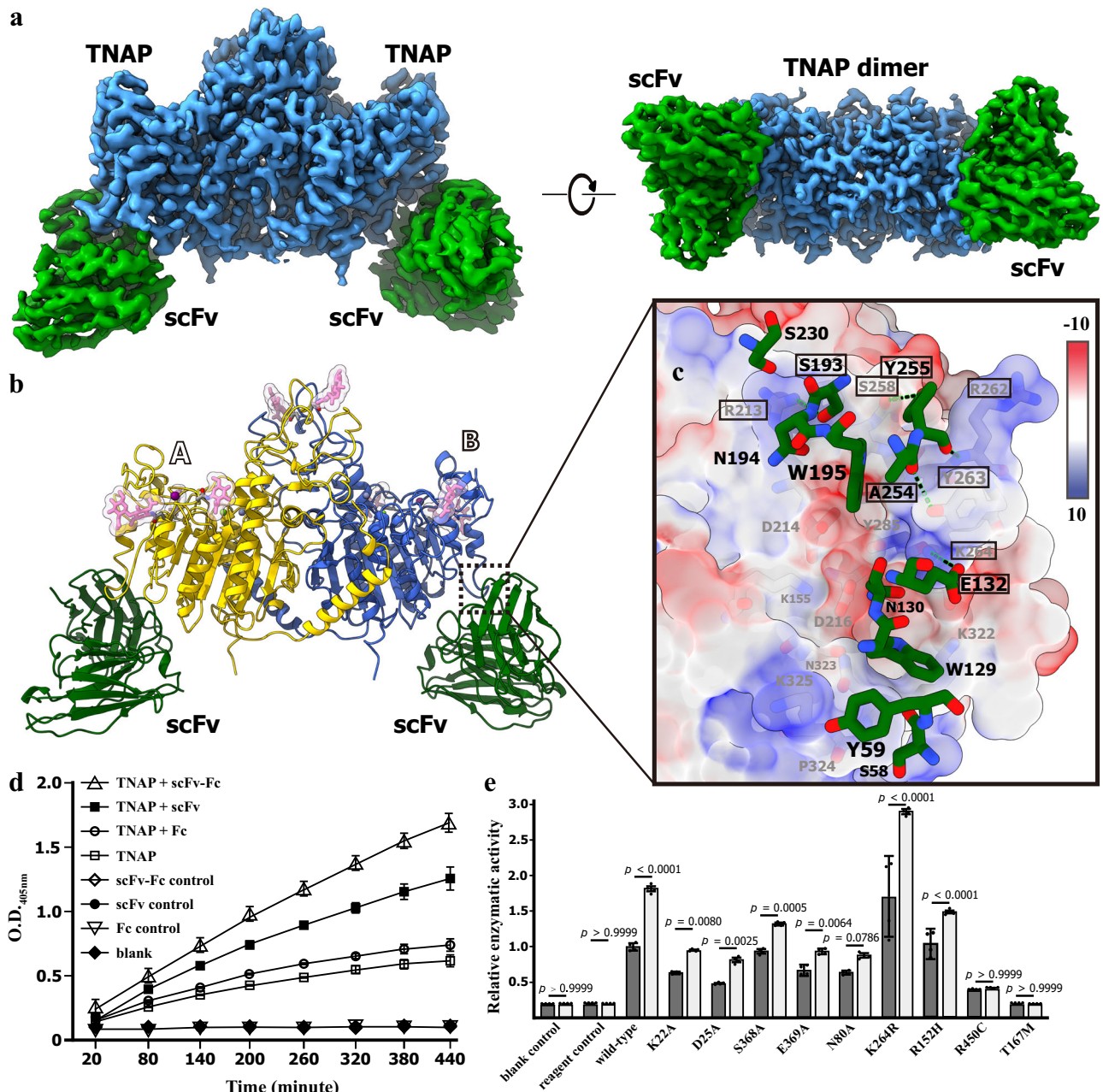

**Fig. 4 | The hTNAP$^{18-500}$-scFv complex. a** The cryo-EM map for hTNAP-scFv complex viewed at two angles. The electron density map of TNAP dimer was colored in blue and the electron density map of scFv was colored in green. **b** The cartoon representation of the TNAP-scFv complex, where the dimeric TNAPs were colored in yellow and blue for protomer A and B, respectively, with two scFv molecules in green. **c** The epitope mapping on hTNAP$^{18-500}$. The protomer B was shown as calculated solvent-accessible electrostatic surface-potential maps and the interacting residues from scFv was shown as stick model colored by elements. The critical residues mediating scFv binding were indicated with red frames. **d** The scFv and scFv-Fc (JTALP001) antibody could increase the alkaline phosphatase activities of hTNAP$^{18-500}$ protein. optical density (OD). $n = 3$ biologically independent samples. **e** The scFv-Fc (JTALP001) antibody showed varing effect of hTNAP$^{18-500}$ mutants on alkaline phosphatase activities. The mutants of hTNAP$^{18-500}$ protein were overexpressed in HEK293T and the phosphatase activities in the cell media were measured and calibrated with the expression level as determined by the Western blotting. $n = 4$ biologically independent samples. All data in this figure are represented as mean ± SD. One-way ANOVA with Tukey's multiple comparisons test for (**e**). All experiments were repeated three times independently with similar results. Source data are provided as a Source Data file.

TNAP$^{T167M}$ proteins did not induce osteogenesis, further administration of JTALP001 based on those two mutants failed to induce osteoblast mineralization (Fig. 6e–h). Taken together, we demonstrated that recombinant TNAP protein could enhance osteoblast mineralization, while TNAP$^{R450C}$ or TNAP$^{T167M}$ mutants had no such effect. The TNAP agonist antibody JTALP001 is potent in activating the biomineralization of the functional TNAP mutants.

## Discussion

The amino acid sequences of hTNAP and hPLAP exhibit approximately 55% identity (Supplementary Fig. 2a), resulting in high similarity in their overall folding and metal-binding/catalytic sites. Our analysis reveals that the dimeric forms of hTNAP and hPLAP share a similar architecture, with an overall RMSD of 1.36 Å. The calcium-binding and catalytic sites also show the highest degree of similarity

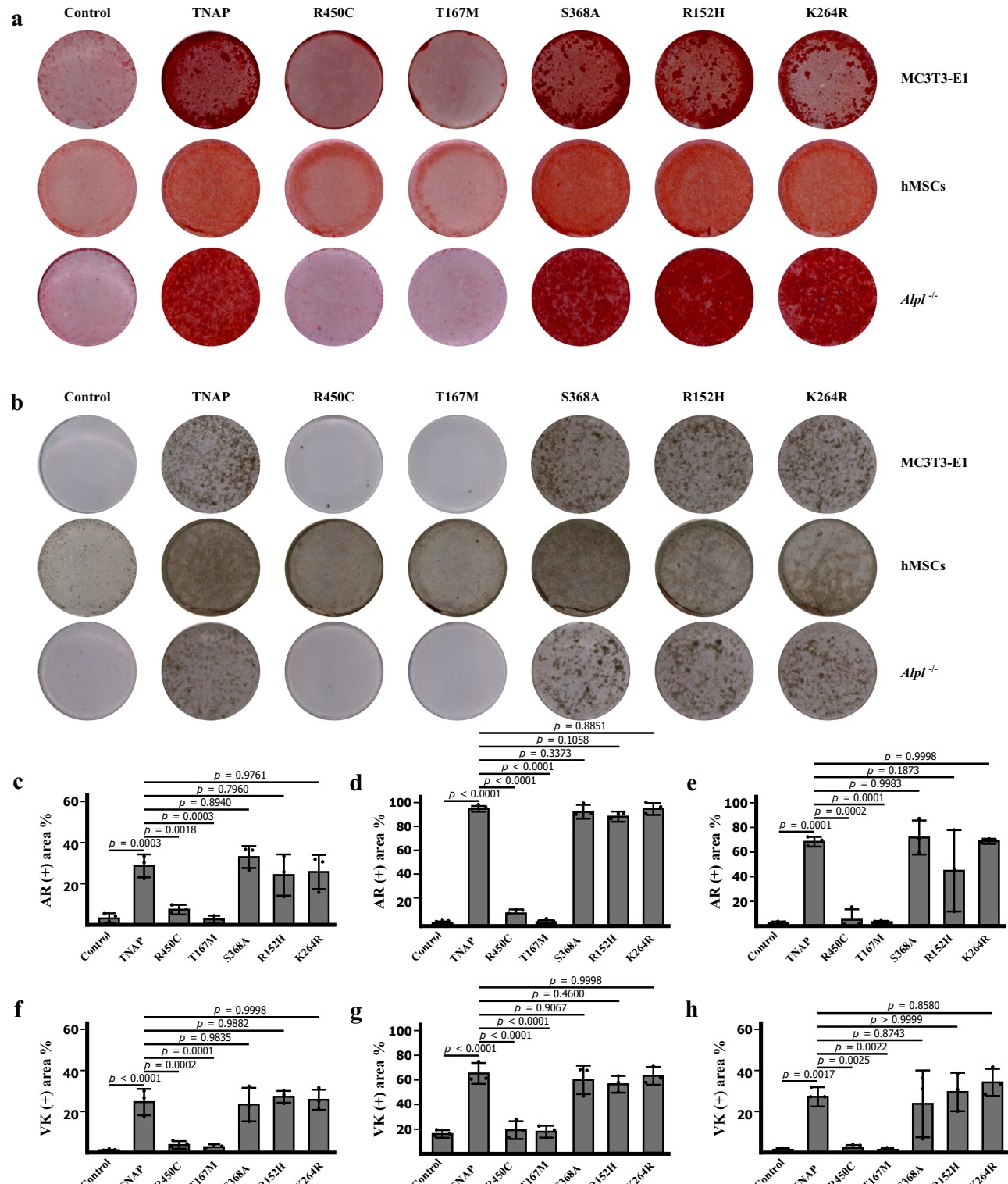

**Fig. 5 | TNAP, TNAP^S368A, TNAP^R152H, TNAP^K264R but not TNAP^R450C or TNAP^TI67M enhanced osteoblast mineralization. a**, **b** Alizarin red staining and Von Kossa staining of MC3T3-E1 cell after differentiation, hMSCs and Alpl^−/− pre-osteoblasts with the administration of wild type TNAP, TNAP^R450C, TNAP^TI67M, TNAP ^S368A, TNAP^R152H, or TNAP^K264R protein. **c**–**e** Quantification of the percentage of Alizarin red (+) area of MC3T3-E1 cell, hMSCs and Alpl^−/− pre-osteoblasts. $n = 3$ biologically independent samples. **f**–**h** Quantification of the percentage of Von Kossa (+) area of MC3T3-E1 cell, hMSCs and Alpl^−/− pre-osteoblasts. $n = 3$ biologically independent samples. All data in this figure are represented as mean ± SD. One-way ANOVA with Tukey's multiple comparisons test for (**c**–**h**). All experiments were repeated three times independently with similar results. Source data are provided as a Source Data file.

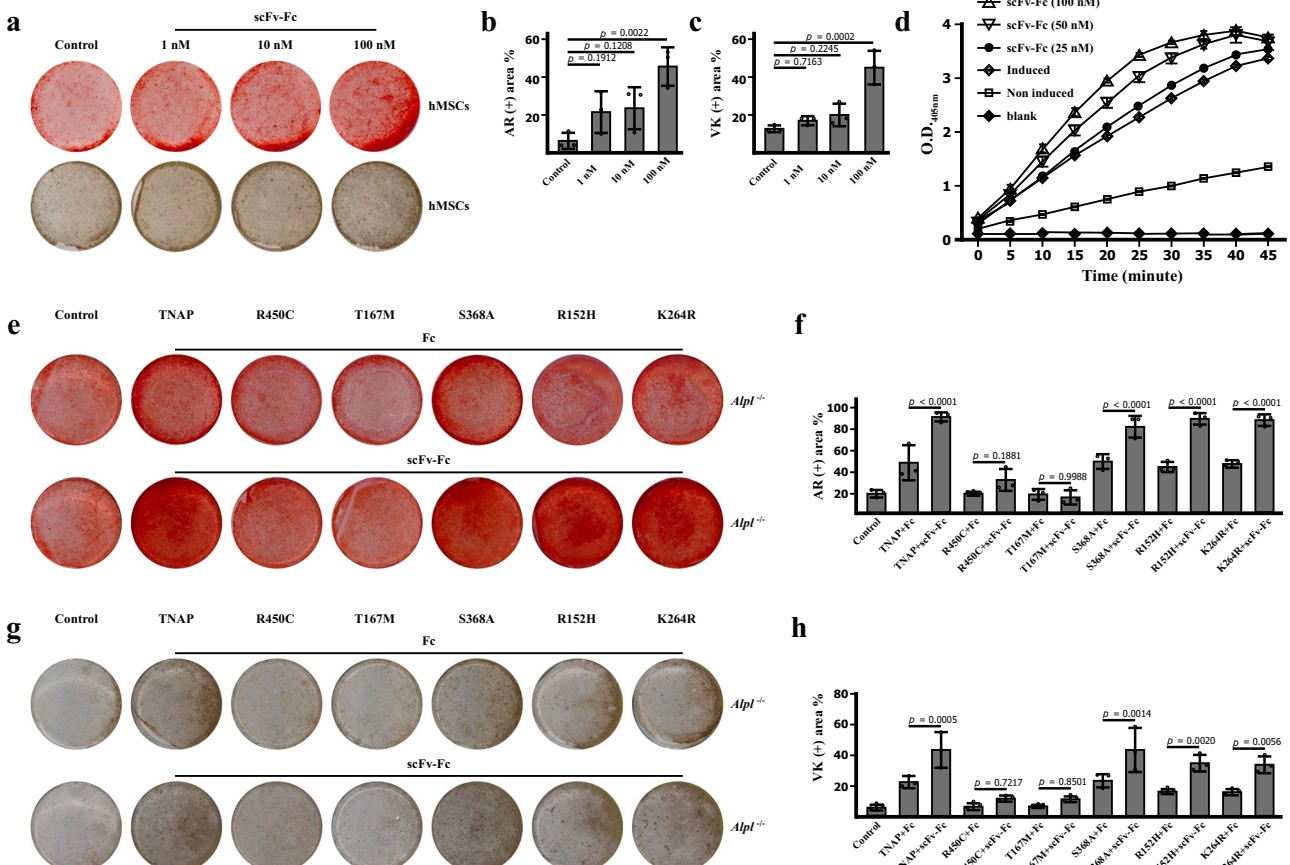

**Fig. 6 | JTALP001 antibody promoted osteoblast mineralization. a–c** scFv-Fc (JTALP001) promoted hMSCs mineralization. Alizarin red staining and Von Kossa staining of hMSCs at different concentrations of JTALP001 after 14 days of osteogenic induction (left). Quantification of the percentage of Alizarin red (+) area and Von Kossa (+) area (right). n = 3 biologically independent samples. **d** The phosphatase activity of cell lysates from hMSCs treated with osteogenic medium for 7 days measured by phosphatase substrate assay. n = 3 biologically independent samples. **e–h** JTALP001 boosted TNAP induced osteoblast mineralization. Alizarin

red staining and Von Kossa staining after differentiation of $Alpl^{-/-}$ pre-osteoblasts for 21 days in the presence of wild TNAP, $TNAP^{S368A}$, $TNAP^{R152H}$, $TNAP^{K264R}$, $TNAP^{R450C}$, or $TNAP^{T167M}$ proteins with further administration of 100 nM Fc fragment as control or 100 nM scFv-Fc (JTALP001). Quantification of the percentage of Alizarin red (+) area and Von Kossa (+) area. n = 3 biologically independent samples. All data in this figure are represented as mean ± SD. One-way ANOVA with Tukey's multiple comparisons test for (**b**, **c**, **f**, **h**). All experiments were repeated three times independently with similar results. Source data are provided as a Source Data file.

(Supplementary Fig. 2b). However, key variants were identified within the oligomeric interfaces of hTNAP, particularly those involving reversed charged or absent residues in hPLAP. For instance, the crucial interacting residues in the interface S, R213, R262, and R263 are absent in hPLAP, and negatively charged residue D156 is replaced by positively charged K138 in hPLAP (Supplementary Fig. 2c). Similarly, interface L residues K27 and D31 are substituted by their reversed charged counterparts, D10 and R14, respectively, in PLAP, and H281 is absent in hPLAP. These differences likely contribute to the higher oligomerization tendency of hTNAP. Moreover, R450, a residue related to hypophosphatasia in hTNAP, is substituted with D428 in PLAP, and its interacting residue in the neighboring subunit of the dimer, D404, is absent in hPLAP. In animal alkaline phosphatases such as TNAP and PLAP, the N-terminal helix, Ca²⁺-binding site, and crown-like domain are major features absent in bacterial alkaline phosphatases. These features are well-characterized in hPLAP's structural work and homology modeling of hTNAP by refs. [14,17]. The crown-like domain has been suggested to play a role in dimerization, which was confirmed by the hTNAP structures. Additionally, the calcium-binding site represents a feature that is absent in bacterial alkaline phosphatases, which might have evolved in response to the skeletal mineralization needs of animals. This feature helps TNAP to increase its activity in producing phosphate when calcium ions are available, thereby enabling more efficient calcium phosphate-based mineral deposition. We also

observed that the N-terminal helix, which is involved in PLAP dimerization, mediates oligomerization in TNAP by forming a series of electrostatic and hydrogen bonds with helices H3 and H18 of other TNAP dimers (Fig. 3e). This suggests an additional function of the N-terminal helix in TNAPs, beyond that observed in PLAP.

The crystallographic analysis of hTNAP has uncovered an unexpected octameric structure that has not been observed in other previously reported AP structures. The formation of an octamer from the tetramerization of dimeric hTNAP leads to the formation of a large AP machinery composed of eight functional subunits. The functional significance of octamerization in hTNAP is not fully understood. However, the disruption of the hTNAP function resulting from mutations on the octameric interface implies that hTNAP may be stabilized by octamerization, which prevents the interfaces from exposure. This stabilizing effect could be important for TNAP to maintain a long-term biological function, especially when it enters the circulation to reach other tissues or organs. This speculation is supported by the finding that the TNAP agonist antibody JTALP001 partially mimics the octameric effects by binding with and occupying the interface S of hTNAP. Nevertheless, future dedicated efforts for in vivo analysis of the function of the octameric hTNAP in osteogenesis are clearly warranted.

TNAP is a glycosylphosphatidylinositol (GPI)-anchored protein that exists in two major forms: a membrane-bound form with the GPI anchor and a soluble form following cleavage of the anchor by

phosphatidylinositol-specific phospholipase C (PI-PLC)[38,44,45]. The spatial conflict between the membrane and neighboring TNAP subunits prevents the formation of an octamer with more than one GPI anchor. Therefore, the membrane-bound form is dominated by dimeric TNAP (Supplementary Fig. 10). However, once released into circulation upon the PIPLC cleavage, TNAP can polymerize into its oligomeric form. Our structural findings on the dimeric-oligomeric form transition are supported by biochemical studies by Nosjean et al., which showed that the presence of a GPI anchor inhibited the oligomerization of TNAP, and oligomerization only occurred in the GPI-free form[38]. The physiological relevance of the TNAP octamer requires further investigation.

HPP is classified into six subtypes, including prenatal, perinatal, infantile, childhood, adult benign, and odontohypophosphatasia[23,46]. This classification is mainly based on the clinical severity and ages of onset. However, the relationship between clinical severity and the mutation sites has yet to be precisely elucidated. Especially, the pathology of the non-active site mutation inducing HPP remains unexplored. Here, we determined the structure of human TNAP and were able to present a structure-based molecular subtyping strategy. The severity of the disease might depend on the locations of the pathological mutations, such as the catalytic center of the enzyme, the metal ion binding sites, the dimeric interface, and the octameric interface. For example, the R450C mutation in the crown region is to disrupt dimerization and the T167M mutation in the catalytic center is destructive to the enzymatic activity. Consistently, the clinical manifestations of the patients with R450C mutation caused the prenatal lethal form of HPP, while T167M mutation led to severe childhood form[28,47]. Our AP assay and osteoblast mineralization assay confirmed the genotype-phenotype correlation. On the other hand, mutations such as K264R maintaining high AP activity is associated with the adult benign form of HPP[48]. Therefore, our crystal structure results provide a precise map for molecular subtyping of HPP.

At present, enzyme replacement therapy has been proved to be efficient for the treatment of HPP. The recombinant fusion protein asfotase alfa (trade name STRENSIQ®) is composed of human TNAP non-membrane-binding domains with Fc fragments and polyarginine motifs[10]. Like the other enzyme replacement therapies (ERT), the efficacy could be reduced by the anti-enzyme antibody[49]. So the enzyme replacement also needs alternative candidates. Different from the recombinant protein therapy, the TNAP agonist antibody JTALP001 was identified capable of enhancing the TNAP functions by binding human TNAP to stabilize its active form. Moreover, JTALP001 could also enhance the AP activity of mutant TNAPs carrying pathological mutations at the octameric interfaces, and thus shows further therapeutic potentials for HPPs associated with those mutations. By contrast, the JTALP001 failed to rescue the AP activity of the TNAP$^{R450C}$ or TNAP$^{T167M}$ mutants, indicating the JTALP001 functions more efficiently on the destabilizing mutants, instead of the deactivated mutans. Since decreased TNAP activity is not only associated with HPP, but also crucial in aging patients with decreased bone formation[50], the therapeutic potential of this TNAP-agonist antibody could be extended to the treatment of bone defects or fractures in those patients.

## Methods

### Ethics statement

The research in this study complies with all relevant ethical regulations. All the experiments for the study were performed following standards according to the protocol written and approved by the Ethics Committee of Shanghai Ninth Peoples Hospital (Shanghai, China).

### Protein expression and purification

The human *ALPL* gene (UniProt ID P05186) was synthesized from WZ Biosciences Inc., China and cDNA encoding the amino acid 18–500 was subcloned into a modified pFast-Bac Dual vector (Invitrogen, USA)

with a C-terminal TEV protease recognition sequence followed by a Flag tag and 8× polyhistidine tag, as well as an HA signaling peptide at N-terminal. The recombinant baculovirus encoding the *ALPL* gene was generated and infected the *Trichoplusia ni* High Five cells line (Hi5) at the density of $2.5 \times 10^6$ cells/milliliter. The Hi5 cells line (Thermo Fisher Scientific, B85502) were cultured in ESF921 medium (Expression Systems, USA) at 27 °C, 120 rpm for the overexpression of hTNAP$^{18–500}$. The Hi5 cells were harvested 96 h after infection for protein purification.

For the overexpression in the mammalian cells, the human wild-type *ALPL* gene (amino acid 18–500) and mutants were similarly cloned into a modified pcDNA3.4 vector with a Pre-Scission Protease recognition sequence followed by a Flag tag and 8× polyhistidine tag at C-terminal, as well as an HA signaling peptide at N-terminal. Expi293F cells (Thermo Fisher Scientific, A14527) were cultured in a chemically defined Union-293 medium (Union-Biotech, China) at 37 °C, 120 rpm, supplied with 5% $CO_2$. When cell density reached $2.5 \times 10^6$ cells/milliliter, the cells were transiently transfected with the pcDNA3.4-based expression plasmids using PEI MAX (Polysciences, USA). The cells were harvested 96 h after transfection for protein purification. All of the primers were listed in supplementary primer list table.

For the purification of TNAP proteins, the supernatants of the cell cultures were collected by centrifugation and then subjected to the affinity chromatography with Nickel-chelating resin (Smart-Lifesciences, China). After eluted with a buffer comprised with 20 mM HEPES (pH 7.5), 150 mM NaCl, 10% glycerol, and 300 mM imidazole, the eluates were concentrated and the proteins of interest were further purified with size-exclusion chromatography using Superdex 200 Increase 10/300 GL (GE Healthcare, USA) column pre-equalized in a buffer comprised with 20 mM HEPES (pH 7.5), 150 mM NaCl. The fractions of the UV absorbance peaks were pooled and the purities were examined with SDS-polyacrylamide gel electrophoresis (SDS-PAGE).

### Crystallization

The human TNAP$^{18–500}$ crystals were initially grown in an under-oil configuration where 1 μl of the protein solution was mixed with an equal volume of crystallization solution covered by mineral oil layer. The TNAP crystals were obtained in 20% (w/v) PEG3350, 640 mM sodium acetate, pH5.0. Before flash-freezing in liquid nitrogen, the crystals were cryoprotected by soaking for 2–5 s in crystallization solution supplemented with 20% (v/v) ethylene glycol. After further crystallization trial, the more TNAP crystals were grown by vapor diffusion in sitting-drop configuration where 1-2 μl of protein solution was mixed with an equal volume of crystallization solution comprising of 17% (w/v) PEG4000, 30 mM Tris, pH8.0, and 100 mM MgCl$_2$. Before flash-freezing in liquid nitrogen, the crystals were cryoprotected by soaking for 2–5 s in crystallization solution supplemented with 30% (v/v) ethylene glycol.

### X-ray data collection and structure solution

X-ray Diffraction data were collected at the Shanghai Synchrotron Radiation Facility (SSRF) beamlines BL18U1 and BL19U1 and processed with HKL2000. The initial model was solved by molecular replacement with Phaser in Phenix (version 1.18.2, Phenix Project, USA) using the human PLAP structure (PDB ID 1EW2) as the starting model. Iterative refinement cycles were carried out using Phenix, Refmac5 (version 7.1, from CCP4 program suite, UK) and Coot (version 0.8.9.2, Crystallographic Object-Oriented Toolkit, UK). The crystallographic parameters and data collection statistics are given in Supplementary Table 1.

### Selection, expression, and purification of hTNAP$^{18–500}$ antibody

The hTNAP$^{18–500}$-specific single-chain fragment variable (scFv) antibodies were selected from a combinatorial human monoclonal scFv

antibody phage library. The scFv phage library was generated from peripheral blood mononuclear cells (PBMC) of 15 donors as described before[51]. The purified recombinant TNAP protein was coated in a 96-well immunosorp plate (NUNC MaxiSorb, USA) as bait. Phagemids, i.e., displaying the antibody library, were first incubated in the TNAP coated well. The unbound phages were removed by washing three times with phosphate-buffered saline (PBS) and PBST (PBS supplemented with 0.1% Tween-20), respectively. The TNAP-bound phages were eluted using an eluting buffer (glycine-HCl, pH 2.2) and infected XL-1 blue bacteria, which were grown at 37 °C on LB agar plates. Phages were collected from the plate and amplified for the next round of selection. The panning procedure were iterated three times. The positive colonies were selected and analyzed by phage ELISA. The positive clones were sequenced using Sanger sequencing.

DNA sequences encoding the variable regions of the enriched antibody was cloned into a pFuse-Fc expression vector (Invitrogen, USA). ScFv-Fc (JTALP001) antibody was expressed in Expi293F cells for 4 days and further purified by protein A column (Hitrap Protein A HP, USA). For crystallization, the Fc fragment was replaced by His-tag, generating the scFv construct. The scFv antibody was also expressed in Expi293F cells for 4 days, and then purified by Nickel-chelating resin (Smart-Lifesciences, China).

### Preparation of hTNAP¹⁸⁻⁵⁰⁰-scFv complex

The hTNAP^18–500 protein purified from Hi5 cells was incubated with scFv in a molar ratio of 2.5:1 for 30 min at 4 °C and the mixture was subjected to size exclusion chromatography for separation. The fractions were pooled and the formation of the complex was verified by SDS-PAGE.

### cryo-EM sample preparation, data collection, and processing

The glow discharged holey carbon grids (Quantifoil R1.2/1.3 Au, 300 mesh) were covered with three-microliter aliquots of the protein sample for 40 s, and blot 1–2.5 s, and then rapidly plunged into liquid ethane cooled by liquid nitrogen. This procedure was operated at 8 °C and 100% humidity by using a Vitrobot Mark IV (FEI). Utilize a Talos Arctica (FEI) at 200 kV to image the grids, and the images were collected automatically with EPU software (version 2.12.0.2771REL) on a Falcon 4 camera operated in count mode.

Furthermore, the Titan Krios transmission electron microscope (FEI) was used to image the grids at 300 kV, and the K3 camera (Gatan) was also operated in count mode. A GIF Quantum energy filter was placed at the end, which functioned in zero–energy loss mode with a slit width of 15 eV. Data were collected at a nominal magnification of 105,000× (corresponding to a physical pixel size of 0.86 Å), with a defocus range between −1.0 and −2.5 μm. The dose rate was set to -18.2 electrons/Å$^2$ pers, and the total exposure time was 2.7 s, resulting in a total dose of 50 electrons/Å$^2$, fractionated into 32 frames.

All the data processing steps were performed using program RELION-3.1.1 and cryoSPARC (version 3.2.2)[52,53]. A total of 8391 cryo-EM (300 kV) images were collected, and motion correction was performed using MotionCorr2-1.1.0 with dose weighting on the dose-fractioned image stacks. The contrast transfer function (CTF) parameters of each image were determined, and automatic particle picking was carried out by using Gautomatch v0.56 (www2.mrc-lmb.cam.ac.uk/research/locally developed-software/zhang-software/#gauto) based on templates generated from a 2D reference from 200 kV EM data. For TNAP-scFv complex, 11,297,222 particles were firstly extracted with 2× binning, the 2D classifications and Heterogeneous Refinement excluded particles with poor quality. 2,090,232 particles were then re-extracted with 2× binning and subjected to the global 3D classification. 556,495 particles were selected and the best EM map was setting low-passed to 20 Å, then all four references generated using Ab-initio Reconstruction were combined to generate a multi reference for further 3D classification. After two rounds of 3D classification, 324,279 particles

corresponding to appropriate classes were re-extracted without binning and subjected to process with cryoSPARC. Then, these particles were subjected to Ab-initio Reconstruction and five references were generated. All particles were pooled into Heterogeneous Refinement, and after seven cycles of classification, 277,515 particles belonged to the best class, which were subjected into Bayesian polishing, Non-uniform Refinement (Legacy) with C2 symmetry and an adaptive solvent mask, thereby yielding a map with an overall resolution of 2.96 Å. Local resolution estimation was performed by cryoSPARC and all the resolutions were estimated by the gold-standard Fourier shell correlation 0.143 criteria with the high-resolution noise substitution[54,55]. All the processing procedures were illustrated in Supplementary Fig. 15.

### Cryo-EM structure model building and refinement

The crystal structures of antibody 2D10 scFv fragment (PDB ID 5WN9) and the dimeric TNAP crystal structure were combined and used as reference model for TNAP-scFv initial model building using Phenix, followed by iterative manual adjustment using Coot and Real-space refinement in Phenix to determine the cryo-EM structural model for TNAP-scFv (Supplementary Figs. 16 and 17). The parameters for cryo-EM data collection, processing and structure refinement are given in Supplementary Table 2.

### Cells culture and transfection

HEK293T cells (NCACC, SCSP-502) were adherent cultured in Dulbecco's Modified Eagle's Medium (DMEM; Hyclone, USA) with 10% fetal bovine serum (FBS; Hyclone, USA). Cells in 24-well plates were transfected with pcDNA3.4-based expression plasmids carrying cDNAs encoding wild-type TNAP or its mutants as above-mentioned. After 48 h, the supernatants and cells in the culture were harvested for western blot and the enzymatic activity assay.

MC3T3-E1 cell lines were obtained from the American Type Culture Collection (ATCC, CRL-2593) and maintained in alpha-modified Eagle's medium (α-MEM; Hyclone, USA) with 10% FBS and 1% Penicillin-Streptomycin (Gibco, USA).

Mouse pre-osteoblasts were isolated from calvarial bones. Bones were digested with 0.3% collagenase (Sigma, Japan) for 2 h, and terminated by α-MEM supplemented with 10% FBS. Then, the cells were resuspended and cultured in α-MEM supplemented with 10% FBS, 2 mM L-glutamine and 1% Penicillin-Streptomycin. The adherent cells were cultured for 3–4 days and passaged at 90% confluence. Early passage (P3-P5) of pre-osteoblasts were used for subsequent study.

Human mesenchymal stem cells (hMSCs) were obtained from 3 female patients (age 50–61 years) undergoing routine total hip replacement surgery at the Ninth People's Hospital (Shanghai, China). All patients gave written, informed consent to donate biological material for research purposes. The hMSCs were cultured in α-MEM supplemented with 10% FBS, 2 mM L-glutamine, and 1% Penicillin-Streptomycin. Early passage (P3-P5) of hMSCs were induced of osteogenic differentiation. The ethics for harvesting human mesenchymal stem cells was also approved by the Ethics Committee of the Ninth People's Hospital, Shanghai Jiaotong University School of Medicine (Ethics Number: SH9H-2022-TK36-1).

### Alkaline phosphatase (ALP) activity assays and staining

The assay used p-nitrophenyl phosphate (pNPP; Sangon Biotechnology, China) as the phosphatase substrate which could be enzymatically hydrolyzed into a yellow-colored product (maximal absorbance at 405 nm). The reactions were started by adding 50 μl substrate solution comprising of 10 mM pNPP to each well of a 96-well microtiter plate which was pre-loaded with 50 μl of sample solution. The absorbance of 405 nm was measured at 37 °C by using Infinite M200 Pro NanoQuant (TECAN, Switzerland).

Alkaline phosphatase staining of the cells was performed as followings: The hMSCs were washed twice with PBS and fixed with 4%

paraformaldehyde (PFA; Beyotime, China). Then, cells were stained with the Alkaline Phosphatase Color Development Kit (Beyotime, China) according to the manufacturer's protocol. The stained cells were observed by the scanner (ESPON, Japan).

## Osteogenic differentiation assays

MC3T3-E1 cells, mouse pre-osteoblasts or hMSCs were plated in 48-well plates (Corning, Japan) at density of 30,000 cells per well. On the second day, cells were induced by osteogenesis induction media including α-MEM, 10% FBS, 100 nM dexamethasone (Sigma, Japan), 50 μM ascorbic acid (Sigma, Japan) and 4 mM β-glycerophosphate (Sigma, Japan). Wild type TNAP, TNAP mutants, Fc or scFv-Fc was added directly to the osteogenesis induction media. Medium was replaced with fresh one every third day. Alizarin red staining and Von Kossa staining were conducted on day 14 or day 21. Cells were fixed with 4% PFA and stained with Alizarin Red S Solution (OriCell, USA) or Von Kossa Staining Kit (GENMED, USA) to assess osteogenesis differentiation. Pictures were captured using a scanner at 1200dpi resolution (ESPON, Japan).

## Western blot

The wild-type and mutations of hTNAP[18–500] proteins were over-expressed by transfected with the pcDNA3.4-based expression plasmids in HEK293T. After 48–60 h, the cell supernants and cell lysates in RIPA lysis buffer (Beyotime, China) were collected respectively. Mouse pre-osteoblasts were also harvested in RIPA lysis buffer. The samples were subjected to separate on 4–20% polyacrylamide gel (GenScript, China), transferred to PVDF membranes (Millipore, USA), and blocked with 5% milk. Then, the membranes were incubated with primary antibody, mouse ALPL antibody (R&D Systems, Cat no: AF2910, Lot: WYM0221111, 1:200) or beta actin Mouse monoclonal antibody (Affinity, Cat no: T0022, Lot: 54o2802, 1:1000) overnight at 4 °C and bands were revealed after incubation with secondary antibodys including Anti-mouse IgG, HRP-linked Antibody (CST, Cat no: 7076P2, Lot: 36, 1:5000), Anti-rabbit IgG, HRP-linked Antibody (CST, Cat no: 7074P2, Lot: 25, 1:5000), Anti-mouse IgG (H + L) (DyLight™ 800 4X PEG Conjugate) (CST, Cat no: 5257, Lot: 11, 1:5000) and Rabbit Anti-Goat IgG H&L (HRP) (Abcam, Cat no: ab6741, Lot: GR3324245-8, 1:5000). Additionally, the overexpressed hTNAP[18–500] proteins and the protein mutants were incubated with HRP-conjugated DYKDDDDK Tag Monoclonal antibody (ProteinTech, Cat no: HRP-66008, Clone no: 8H6A10, Lot: 21006202, 1:10000).

## Mice

*Alpl*[+/−] mice were kindly gifted by Prof. Wenjia Liu from the Second Affiliated Hospital of Xi'an Jiaotong University, Xi'an, China. Heterozygous were mated to breed *Alpl*[−/−] mice. Wild type mice were purchased from Weitonglihua Corporation (Beijing, China). All mice were maintained on C57BL/6 background. Primers used for genotype indentification: PCR-i, T014130-F1: CATTGTCAGCTCCAGA-GATGGAAC, T014130-R1: GACCTGAGCGTTGGTGTTATATG; PCR-ii, T014130-F2: GACCTGAGCGTTGGTGTTATATG, T014130-R2: CATCTCCCAGGAACATGATGAC. All mice were used for analysis regardless of sex. All mice were housed in pathogen-free conditions with constant ambient temperature (22 ± 2 °C) and humidity (55 ± 10%), with an alternating 12-h light/dark cycle. All mice were euthanized by isoflurane followed by cervical dislocation. All of the procedures that involved animals were approved by the Institutional Animal Care and Ethics Committee of the Ninth People's Hospital, Shanghai Jiaotong University School of Medicine (Shanghai, China; Ethics Number: SH9H-2022-A037-SB).

## Blue native PAGE

The hTNAP[18–500] proteins purified from Hi5 cells were used for Blue native-PAGE analysis with the procedure as described[56]. In brief, the protein samples were mixed with native-PAGE sample loading buffer (0.1% Ponceau S, 50% Glycerol) and subjected to the native-PAGE for 2.5 h. The proteins in the navtive gel were visualized by Coomassie-blue R250 staining.

## Chemical cross-linking assay

The hTNAP[18–500] proteins purified from Hi5 cells were used for chemical cross-linking assay. The cross-linking reactions were performed in room temprature for half an hour by mixing the TNAP[18–500] proteins with the crosslinking reagents in a final concentration of 4 mM as following: glutaraldehyde (GA, from Sangon Biotech, China), dimethyl pimelimidate (DMP, from Thermo, USA), sulfosuccinimidyl 6-(3′-(2-pyridyldithio)propionamido)hexanoate (Sulfo-LC-SPDP, from Thermo, USA). The crosslinking reactions were quenched with 50 mM Tris-HCl, pH 7.5 and the reaction mix were then subjected to the SDS-PAGE analysis.

## Fluorescence-activated cell sorting (FACS)

In the flow-cytometry binding experiment, the full-length hTNAP (uniprot No. P05186), fused with P2A-EGFP, was transiently transfected into HEK293T cells. The HEK293T cell expressing GFP was used as a negative control. After 1 day cultivation, cells were collected and re-suspended in an ice-cold FACS buffer, containing PBS, 0.05% BSA and 2 mM EDTA. The TNAP protein expressing cells were then incubated with scFv antibody for 20 min on ice, and washed with 1 ml ice-cold FACS buffer, spun, and re-suspended in a 100 μl ice-cold FACS buffer containing the Alexa 633 conjugated goat anti-human IgG secondary antibody (A21091, Thermo). After incubating on ice for 15 min, the cells were washed twice and re-suspended in a FACS buffer. The harvested cells were sorted on a flow cytometer (CytoFLEX S, Beckman Culter) to determine relative binding level by the antibodies to the cell surface TNAP. The HEK293T cells were used to set the gates. The FACS data were analyzed using FlowJo v10 software.

## The static light scattering (SLS) analysis of hTNAP

The static light scattering (SLS) experiments were performed using an Agilent 1260 Infinity Isocratic Liquid Chromatography System coupled with a Wyatt Dawn Heleos II Multi-Angle Light Scattering (MALS) detector (Wyatt Technology) and a Wyatt Optilab T-rEX Refractive Index Detector (Wyatt Technology). 40 μL samples were loaded onto the WTC-030S5 column (Wyatt Technology) with a mobile phase containing 150 mM NaCl, 20 mM HEPES, pH 7.5 at a flow rate of 0.5 mL min⁻¹. BSA (ThermoFisher Scientific) was used to calibrate the static light scattering system. Data were recorded and processed using Astra 7 software (Wyatt Technology).

## Reporting summary

Further information on research design is available in the Nature Portfolio Reporting Summary linked to this article.

## Data availability

The coordinates are deposited at Protein Data Bank with accession code: 7YIV, 7YIW, and 7YIX. The coordinates used in structural solving are available at Protein Data Bank with accession codes: 1EW2 and 5WN9. The cryo-EM maps have been deposited in the Electron Microscopy Data Bank with accession codes EMD-33865. Source data are provided with this paper.

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

## Acknowledgements

The authors thank Drs. Ming Lei, Jie Zhao, and Lijun Wang (for scientific discussion); Yijun Gu, Rijing Liao, Shufang He and Mi Cao (for assistance with the data analysis). This work was supported by National Natural Science Foundation of China (82272519 and 82072468, Y.C.; 92068102 and 81772373 A.Q.), National Key Research and Development Program of China (2018YFC1004704, Y.C.), Shanghai Frontiers Science Center of Degeneration and Regeneration in Skeletal System, the Shanghai Science and Technology Committee (20S11902000 Y.C.; 23ZR1437600 P.M.), SHIPM-pi fund (No. JY201804, Y.C.) from Shanghai Institute of Precision Medicine, the Ninth People's Hospital, Shanghai Jiao Tong University School of Medicine, Shanghai Key Laboratory of Orthopedic Implants (KFKT202207). This work was also supported by Innovative Research Team of High-level Local Universities in Shanghai (SHSMU-ZLCX20211700) and the Special Project of Shanghai Synchrotron Radiation Facility (SSRF) BL18U1 (2020-NFPS-ZD-000146). We thank the staff members of the Electron Microimaging Center, Bioimaging Facility, and Proteomics Platform at Shanghai Institute of Precision Medicine for providing technical support and assistance in data collection. SSRF beamlines BL18U1 and BL19U1 were used for X-ray crystallography data collection.

## Author contributions

Y.C. and A.Q. conceived the study. Y.C., P.M., and A.Q. designed the experiments. Y.Yu, K.R., and X.C. performed the biochemical and cell-based experiments. Y.Yu, Q.Z., Y.X., Y.L., B.R., D.Y., and Y.S. performed structural biology experiments. Y.C. and D.Y. built and refined structural models. Y.Yao and H.X. performed the biochemical assay of antibody. Y.C., A.Q., P.M., Y.Yu, and K.R. wrote the manuscript.

## Competing interests

The authors declare no competing interests.
