## [Peer Review File · Nature Communications]

The structural pathology for hypophosphatasia caused by malfunctional tissue non-specific alkaline phosphataseREVIEWER COMMENTS

Reviewer #1 (Remarks to the Author):

This is a well written paper on an important topic. The introduction provides appropriate background for what studies have previously revealed regarding the structure of AP and clearly explain why their atomic resolution TNAP structure is essential for revealing the true structure of TNAP. This structure allows them to provide a structural basis for how an agonistic antibody could promote TNAP activity. They also provide evidence that their secreted recombinant TNAP (lacking GPI anchor sequence) is functional as TNAP and that TNAP with disease mutations or treated with EGTA to remove Ca^{++} has diminished function in vitro. I will defer comments on the structural analysis to reviewers with expertise in this area, except to say that, if accurate, this structure could provide important information on mechanisms by which different mutations affect TNAP activity, and that may provide for alternative treatments, such as the agonist antibody tested here.

Results also show that administration of TNAP but not two mutant forms of TNAP can rescue mineralization of TNAP deficient cells, and that the agonist antibody can further enhance wild type TNAP activity. It is not clear why they do not test TNAP with mutations located in the interfaces, which they hypothesize are essential for function.

While the authors state that the agonist antibody could be used to rescue AP activity for mutations that occur at structural octameric interfaces, the most disappointing aspect of this manuscript is that they did not test effect of the antibody on this type of TNAP mutation. The reader is therefore left wondering if the antibody could be of use in individuals with hypophosphatasia due to these types of mutations.

Finally, the cell mineralization assays are not performed in an ideal manner. The authors should consider adding inorganic phosphate (NaPO) as opposed to b-glycerophosphate because b-glycerophosphate is a TNAP dependent source of phosphate. As assayed, they are showing how effective TNAP is at hydrolyzing b-glycerophosphate. In addition, alizarin red assays should be supported by Von Kossa assays, as background alizarin staining is not uncommon and can result from cell death and non-HA calcium deposition. Ideally, particularly for the scFv-Fc antibody assays (because the staining is not consistent within the wells), more consistent staining would be achieved to relieve any doubt for the reader as to interpretation. The authors could consider differentiating their cells for 5 days and then adding a phosphate source, which can result in more consistent differentiation and mineralization.

Overall, an interesting and well written manuscript. This reviewer is just left wanting more data supporting efficacy of the "agonist antibody" and its mutation location specificity.

Reviewer #2 (Remarks to the Author):

This paper describes the crystal structure of TNAP at near atomic resolution, contributing a much needed advance in the field of alkaline phosphatase research and hypophosphatasia. The authors should be congratulated for this accomplishment. Having said, there are a number of missed opportunities in describing the structure and the authors go off into 2-3 different directions that pose more questions than answers.

While clearly this is the first time a 3D structure for TNAP has been determined, there is a lot of prior art using homology modeling and site-directed mutagenesis that have served the field well until now. As a reader of this paper, I am wondering: a) how different is the TNAP structure from PLAP; b) has the conclusions of the important role of the N-terminal arm, of the calcium site and of the crown domain largely deduced from mutagenesis experiments been validated by this 3D structure and, if not, in what way has having the structure helped fine-tune the function of those domains. I would have expected to see a long discussion of how the TNAP structure is different or similar to PLAP and also evaluate those earlier studies in the context of this new information. All of that is missing in this paper, and I think the field would benefit from a careful review of the prior art in light of this new structure.

One interesting albeit puzzling revelation from the TNAP structure is that it forms an octamer whereby 8 TNAP dimers seem to oligomerize. The authors then go on assigning some HPP mutations to those novel contact point between adjacent dimers in the octameric structure. First question that comes to mind is: how would an octameric structure function in vivo when TNAP is physiologically a GPI anchored molecule? I would have expected to see at least a modeling of the octamer where each of the monomers in each dimer would have a GPI anchor. Is it even possible? Can the molecule remain anchored? Would it associate with coated pits? How would it affect membrane fluidity in such a large oligomeric complex? As presented, it is unclear if the octameric structure is a feature of the TNAP crystals but not a true reflection of the physiological structure of TNAP in the cell or in vivo.

A similar concern relates to the development of an antibody that apparently enhances catalytic activity. A careful characterization of why/how antibody binding might be affecting the catalytic activity is lacking. From the depiction of the Sc-Fv binding shown in Figure 4 it is apparent that antibody binding would be difficult onto a GPI-anchored TNAP since the antibody seems to attach very close to the C-terminus where the GPI anchor would be covalently attached. Have the authors tested the binding of this antibody to native membrane-bound TNAP? Does it enhance activity under those circumstances?

In short, the authors have some great data here, but there are missing out on an opportunity to describe in full detail a novel structure, referring to and contrasting conclusions from prior art, that is warranted in this context. Furthermore, they are branching into examining TNAP mutations and a novel antibody making a number of claims all of which require further validation. I would suggest that the authors put

together a first paper on the structure, comparing it to PLAP and all prior art that used mutagenesis on TNAP and PLAP to help advance the field. Then perhaps move onto HPP mutations in the context of a second paper where they also have more biological proof that an octameric TNAP structure really exists in vivo. The antibody could be described in that second paper but only after the precise effect on the GPI-anchored TNAP activity had been assessed.

Reviewer #3 (Remarks to the Author):

I have been asked to review the manuscript with particular attention to the structural biology/biochemistry. The X-ray and cryo-EM structures are of high quality. Below are specific comments.

Line 116 and topics related to oligomeric state.

a) Molecular weight standards should be included with the SEC analysis (Fig S1D) to support the claim that TNAP is predominately dimeric in solution. As shown (without such calibration), the SEC analysis does not provide information on oligomeric state. Alternatively, SEC with in-line light scattering, UV, and refractive index measurements can be used to determine the molecular mass of the complex. I find the crosslinking (Fig S1B) a bit inconclusive because glutaraldehyde, which shows the most extensive crosslinking, can sometimes cause non-specific crosslinking between molecules that are not associated with one another.

b) The SEC profile of R450C (in comparison to WT) should be shown to substantiate the supposition that this mutation stabilizes the monomeric form (destabilizes the dimer). Incidentally, why was cystine chosen for this mutation, as it might be expected to cause issues due to disulfide formation? Including the reasoning for the choice of cysteine in the text would help the reader.

c) The SEC profile for T167M should also be shown – it is important to consider whether the protein has a similar fold/oligomeric state to the WT protein. If the T167M mutant reduced protein stability (e.g. caused aggregation), for example, then its lack of ability to promote osteoblast mineralization could be due to misfolding rather than loss of catalytic activity.

Metal binding sites, lines 130 and following. Justification should be given for assigning the metals to Ca/Mg/Zn ions. Were the identities of the metals determined by comparison with previous studies or by anomalous scattering etc.? If by comparison, this could be clarified in the text (with references). If by anomalous scattering, data should be shown. Omit maps are typically not sufficient to distinguish between these ions (because they have similar scattering power).

From the structures, the R450-D404 interaction is not between the side chains, as one would suppose from the text (line 141: “The hTNAP dimer is maintained by the static electric interactions among R71-D458, K99-D90, and R450-D404”). D404 makes a hydrogen bond with the backbone of R450. R450 is exposed to solvent; the side chain of R450 does not interact with D404. Because if the surface exposure of R450, it is unclear (from the structure alone) why the R450C mutation would disrupt dimerization.

From inspection of the structure, R71-D458 forms an ion pair interaction between the side chains, as indicated by the authors. However, K99-D90 does not form an ion pair. Rather, K99 forms a hydrogen bond with the backbone carbonyl oxygen of T85. K99 does not interact with D90. D90 does not form a hydrogen bond with the neighboring subunit. D90 does, however, form an ion pair with R71 from the same subunit – this interaction may stabilize the R71-D458 pair between subunits.

Line 144-149, the “critical role of dimerization in TNAP function” seems to be too strong of a statement. It has not been shown in this study that the indicated mutations disrupt dimerization. The authors suppose that they disrupt dimerization from their locations, but there could be other effects (e.g. expression, folding, reduced catalytic activity, etc.).

Water molecules are not included in the atomic coordinates. The resolutions of the X-ray structures and strong (> 3 sigma) Fo-Fc densities in them would justify their addition to the X-ray structures.

Supplementary Table 1. For Human TNAP18-500, basic pH (PDB ID 7YIV), the RMSD from ideality for bond lengths is listed as 0.54 Å. Perhaps this is a typo.

Average B-factors for the models could be listed in Supplementary Table 1 for protein/ligand.

Reviewer #1 (Remarks to the Author):

This is a well written paper on an important topic. The introduction provides appropriate background for what studies have previously revealed regarding the structure of AP and clearly explain why their atomic resolution TNAP structure is essential for revealing the true structure of TNAP. This structure allows them to provide a structural basis for how an agonistic antibody could promote TNAP activity. They also provide evidence that their secreted recombinant TNAP (lacking GPI anchor sequence) is functional as TNAP and that TNAP with disease mutations or treated with EGTA to remove Ca⁺⁺ has diminished function in vitro. I will defer comments on the structural analysis to reviewers with expertise in this area, except to say that, if accurate, this structure could provide important information on mechanisms by which different mutations affect TNAP activity, and that may provide for alternative treatments, such as the agonist antibody tested here.

Results also show that administration of TNAP but not two mutant forms of TNAP can rescue mineralization of TNAP deficient cells, and that the agonist antibody can further enhance wild type TNAP activity. It is not clear why they do not test TNAP with mutations located in the interfaces, which they hypothesize are essential for function.

We thank the reviewer for the positive comments. According to our summary of pathologically related mutation sites from references and database (**Table S1**), we mapped these pathologically related mutation sites located at the interfaces on the crystal structures of human TNAP. We finally identified dimer interface mutations, including G420A, G420S, R71H, R391H, and R450C. We also identified interface S mutations including D156Y, G491R, C497S, K264R, R152H, I490F, C489S and interface L mutations including Y28D, E311K, T366N, S368del and S368A. Because the dimeric form is the minimal stable state for alkaline phosphatases, and the high hydrophobicity of the dimeric interface makes it very difficult “stabilized” as a monomer, the severe mutations at the dimeric interface usually generate highly unstable AP mutants, which can be expressed in a very low level and hardly purified (**Supplementary Figure 8**). Therefore, we cannot obtain recombinant mutant proteins that exhibit strong destruction to dimers and enzyme activity. We can only get well-expressed recombinant mutant proteins R450C, K264R, R152H, and S368A that appear less destructive to dimerization or enzyme activity.

To examine the effect of different TNAP mutants on osteoblast differentiation and mineralization, we overexpressed and purified the recombinant wild-type TNAP, the dimeric surface mutant TNAP^{R450C}, the active site mutant TNAP^{T167M}, the interface L mutant TNAP^{S368A}, and the interface S mutants TNAP^{R152H} and TNAP^{K264R}. As shown in Figure R1, wild-type TNAP protein, TNAP^{S368A}, TNAP^{R152H}, and TNAP^{K264R} at 600 ng/ml enhanced the mineralization of the MC3T3-E1 cells when compared with the control group. However, the TNAP^{R450C} mutant that located at the crown region and can affect dimerization, failed to induce osteoblast mineralization. Moreover, the TNAP^{T167M} is a stable dimer with no enzyme activity. It failed to induce MC3T3-E1 mineralization. In addition, recombinant wild-type TNAP protein also promoted human bone marrow-derived mesenchymal stem cells differentiation to osteoblasts and subsequent mineralization, whilst TNAP^{R450C} or TNAP^{T167M} exhibited no effect on

osteoblast mineralization.

Figure R1. TNAP, TNAP^{S368A}, TNAP^{R152H}, TNAP^{K264R} but not TNAP^{R450C} or TNAP^{T167M} enhanced osteoblast mineralization. **a-b:** Alizarin red staining and Von Kossa staining after differentiation of MC3T3-E1 cell line, hMSCs and *Alpl*^{-/-} pre-osteoblasts with the administration of the recombinantly expressed and purified proteins of TNAP wild-type, TNAP^{R450C}, TNAP^{T167M}, TNAP^{S368A}, TNAP^{R152H}, and TNAP^{K264R}. **c-e:** Quantification of the percentage of Alizarin red (+) area of MC3T3-E1 cell line, hMSCs and *Alpl*^{-/-} pre-osteoblasts. **f-h:** Quantification of the percentage of Von Kossa (+) area of MC3T3-E1 cell line, hMSCs and *Alpl*^{-/-} pre-osteoblasts. Data are presented as mean ± SD, n = 3. *P < 0.05; **P < 0.01; ***P < 0.001.

While the authors state that the agonist antibody could be used to rescue AP activity for mutations that occur at structural octameric interfaces, the most disappointing aspect of this manuscript is that they did not test effect of the antibody on this type of TNAP mutation. The reader is therefore left wondering if the antibody could be of use in

individuals with hypophosphatasia due to these types of mutations.

We thank the reviewer for the comment. According to our osteogenesis and mineralization assay, we observed that the administration of the recombinantly expressed and purified TNAP^{S368A}, TNAP^{R152H}, and TNAP^{K264R} proteins can partially induce osteoblast mineralization. In addition, administration of JTALP001 further boost osteoblast mineralization when compared with the control group using the Fc fragment. For the TNAP^{R450C}, which is located at the crown region affecting dimerization, the agonist antibody failed to induce osteoblast mineralization. For the TNAP^{T167M} mutant that can form dimer but lose its enzyme activity, further administration of JTALP001 also failed to induce osteoblast mineralization (**Figure R2**). Therefore, we conclude the JTALP001 agonist antibody cannot rescue the mutants which have no enzyme activity or cannot stabilize as a dimer.

Figure R2. JTALP001 antibody promote osteoblast mineralization. **a-c:** scFv-Fc (JTALP001) promoted hMSCs mineralization. Alizarin red staining and Von Kossa staining of hMSCs at different concentrations of JTALP001 after 14 days of osteogenic induction(left). Quantification of the percentage of Alizarin red (+) area and Von Kossa(+) area(right). **d.** The phosphatase activity of cell lysates from hMSCs treated with osteogenic medium for 7 days. **e-h:** JTALP001-boosted TNAP-induced osteoblast mineralization. Alizarin red staining and Von Kossa staining after differentiation of *Alpl*^{-/-} pre-osteoblasts for 21 days in the presence of wild-type TNAP, TNAP^{S368A}, TNAP^{R152H}, TNAP^{K264R}, TNAP^{R450C}, or TNAP^{T167M} proteins with further administration of 100nM Fc fragment control or 100nM scFv-Fc (JTALP001). Quantification of the percentage of Alizarin red (+) area and Von Kossa (+) area. Data are presented as mean \pm SD, n = 3. *P < 0.05; **P < 0.01; ***P < 0.001.

Finally, the cell mineralization assays are not performed in an ideal manner. The authors should consider adding inorganic phosphate (NaPO) as opposed to β -glycerophosphate because β -glycerophosphate is a TNAP dependent source of phosphate. As assayed, they are showing how effective TNAP is at hydrolyzing β -glycerophosphate. In addition, alizarin red assays should be supported by Von Kossa assays, as background alizarin staining is not uncommon and can result from cell death and non-HA calcium deposition. Ideally, particularly for the scFv-Fc antibody assays (because the staining is not consistent within the wells), more consistent staining would be achieved to relieve any doubt for the reader as to interpretation. The authors could consider differentiating their cells for 5 days and then adding a phosphate source, which can result in more consistent differentiation and mineralization.

We appreciate your valuable comments on our osteoblast differentiation assay. We considered adding inorganic phosphate (NaPO) as opposed to β -glycerophosphate during osteogenic differentiation assays. We first explored the concentration of NaPO used for osteogenesis induction and the effect of NaPO. Wild-type and *Alpl*^{-/-} mice-derived pre-osteoblasts were plated in 48-well plates at a density of 3×10^4 cells per well. We differentiated cells for 5 days using 4mM β -glycerophosphate and then adding NaPO or β -glycerophosphate at concentrations of 2 mM and 4 mM for the following osteogenesis induction (**Figure R3**). Alizarin red staining was conducted on day 14. We found that the effect of NaPO on osteoblasts is significantly better than β -glycerophosphate in a concentration-dependent way. No difference is observed in osteoblast mineralization between wild-type and *Alpl*^{-/-} mice-derived pre-osteoblasts by using NaPO (**Figure R3**). Because a 14-day duration is too short to grow mature mineralized nodules by using β -glycerophosphate, we conducted Alizarin red staining on day 21 in this condition and observed less mineralized nodules in *Alpl*^{-/-} mice-derived pre-osteoblasts. Based on these results, we reasoned that NaPO can directly supply inorganic phosphate for the growth of mineralized nodules which is not dependent on the presence of TNAP, but the effect of β -glycerophosphate requires the TNAP function because previous studies have explained that TNAP plays an essential role in skeletal and dental mineralization via its ability to hydrolyze extracellular compounds to increase inorganic phosphate concentration. Considering the purpose of our experiment is to examine the effect of different TNAP mutants on osteoblast differentiation and mineralization, we finally decided to use 4 mM β -glycerophosphate together with 100nM dexamethasone and 50 μ M ascorbic acid for osteogenesis induction.

As required by the reviewer, we conducted alizarin red assays together with Von Kossa assays, and the two methods led to similar results, as shown in Figures R1 and R2 in response to previous comments.

Figure R3. Alizarin red staining of Wild-type and *Alpl*^{-/-} mice-derived pre-osteoblasts. a-c: Alizarin red staining of Wild-type and *Alpl*^{-/-} mice-derived pre-osteoblasts on day 14. 2mM and 4mM NaPO or β-glycerophosphate were used (top). Quantification of the percentage of Alizarin red (+) area (bottom). **d-e:** Alizarin red staining of Wild-type and *Alpl*^{-/-} mice-derived pre-osteoblasts on day 21. 4mM β-glycerophosphate were used (top). Quantification of the percentage of Alizarin red (+) area (bottom). Data are presented as mean ± SD, n = 3. *P < 0.05; **P < 0.01; ***P < 0.001.

Reviewer #2 (Remarks to the Author):

This paper describes the crystal structure of TNAP at near atomic resolution, contributing a much needed advance in the field of alkaline phosphatase research and hypophosphatasia. The authors should be congratulated for this accomplishment. Having said, there are a number of missed opportunities in describing the structure and the authors go off into 2-3 different directions that pose more questions than answers.

While clearly this is the first time a 3D structure for TNAP has been determined, there is a lot of prior art using homology modeling and site-directed mutagenesis that have served the field well until now. As a reader of this paper, I am wondering: a) how different is the TNAP structure from PLAP; b) has the conclusions of the important role of the N-terminal arm, of the calcium site and of the crown domain largely deduced from mutagenesis experiments been validated by this 3D structure and, if not, in what way has having the structure helped fine-tune the function of those domains. I would have expected to see a long discussion of how the TNAP structure is different or similar to PLAP and also evaluate those earlier studies in the context of this new information. All of that is missing in this paper, and I think the field would benefit from a careful review of the prior art in light of this new structure.

We thank the reviewer for the suggestions and comments.

a) We further conducted a comprehensive analysis of the structural comparison between human TNAP and PLAP. There is about 55% identity of amino sequence between TNAP and PLAP, and as a result, they show high similarity in overall folding and metal-binding/catalytic sites. As shown in **Figure R4** below, the dimeric form of TNAP and PLAP are in general same in architecture with an overall RMSD of 1.36 Å, and the highest similarity was observed in their calcium-binding site and catalytic site. Nevertheless, we identified variants within the oligomeric interfaces of TNAP, especially those variant residues with reversed charged or absent in PLAP. For example, the critical interacting residues in the interface S (as shown in **Figure 3d** in the main text), R213, R262, and R263, were absent in PLAP, and negatively charged residue D156 is substituted with positively charged K138 in PLAP. Similarly, the key residues in interface L, K27 and D31, are substituted in PLAP with their reversed charged counterpart, D10 and R14, and H281 is absent in PLAP. These differences likely contribute to the higher oligomerization tendency of TNAP. Additionally, R450 in TNAP, an HPP-related residue at the dimeric interface, is substituted with D428 in PLAP, and its interacting residue in the neighboring subunit of the dimer, D404, is absent in PLAP.

b) The N-terminal arm, Ca²⁺-binding site, and the crown region are major features of animal alkaline phosphatases such as TNAP and PLAP, which are absent in *E. coli* alkaline phosphatase (ECAP). Prior research using homology modeling and site-directed mutagenesis has indeed played an important role in advancing the field's understanding of alkaline phosphatase. While our study is the first to report the crystal structure of TNAP, we agree with the reviewer that it is important to assess how this structure has helped refine our understanding of the N-terminal arm, calcium site, and crown domain. In previous studies, the N-terminal arm, calcium-binding site, and crown region have been well-characterized in PLAP's structural work, as well as in the

homology modeling of TNAP by Le Du et al.^{1,2}. The crown region has been suggested to play a role in dimerization, which was confirmed by our TNAP structures. Additionally, the calcium-binding site represents a novel feature that is absent in bacterial alkaline phosphatase (ECAP), which may have evolved in response to the skeletal mineralization needs of animals. This feature helps TNAP to increase its activity in producing phosphate when calcium ions are available, thereby enabling more efficient calcium phosphate-based mineral deposition. We also found that the N-terminal helix, which is involved in PLAP dimerization, mediates oligomerization in TNAP by forming a series of electrostatic and hydrogen bonds with helices H3 and H18 of other TNAP dimers. This suggests an additional function of the N-terminal helix in TNAPs, beyond that observed in PLAP. In summary, our conclusions regarding the importance of the N-terminal arm, calcium site, and crown domain were drawn from a combination of HPP genetics, mutagenesis, functional assays, and 3D structures of PLAP and TNAP, which mutually support and validate each other.

c)

Figure R4: The structural comparison between crystal structures of human TNAP and PLAP. Upper panel: A structural superposition between hTNAP and hPLAP structures focusing on the calcium-binding site (left) and active site (right). The TNAP and PLAP structures were shown as cartoon models, with ions being shown as spheres and key residues in ion binding and enzymatic activity being shown as a stick model. **Lower panel:** A surface model of hTNAP colored based on the sequence-similarity with hPLAP and viewed from two angles. The residues identical in the sequence alignment of TNAP and PLAP were colored in blue, those sharing high similarities are colored in cyan, and the residues sharing low similarities were colored in gray, except for those showing significant differences from PLAP were highlighted in red and labeled with detailed information.

In addition to the structural comparison between TNAP and PLAP, our analysis of the alignment between our TNAP crystal structure and the homology-modeled TNAP

revealed informative insights. Although the previous model of TNAP provided relatively reliable structural implications due to the high sequence identity between TNAP and PLAP, certain issues needed to be addressed. For example, the R450 (R433 in some literature when numbering without 17-aa signaling peptide) had been proposed as the residue at the entrance of the active site, regulating the approach of the substrate. However, our structure showed that the R450 actually played a role in dimerization in the crown region, interacting with the D404 from the neighboring subunit (**Figure 1f**). Finally, we have incorporated **Figure R4** into our manuscript, which illustrates the structural similarities and differences between TNAP and PLAP, as well as the detailed discussion about the structural similarities and differences between TNAP and PLAP and highlighted them. We thank the reviewer to bring into notice the significance of the structural comparison between PLAP and TNAP, which has greatly improved the quality of our manuscript.

One interesting albeit puzzling revelation from the TNAP structure is that it forms an octamer whereby 8 TNAP dimers seem to oligomerize. The authors then go on assigning some HPP mutations to those novel contact point between adjacent dimers in the octameric structure. First question that comes to mind is: how would an octameric structure function in vivo when TNAP is physiologically a GPI anchored molecule? I would have expected to see at least a modeling of the octamer where each of the monomers in each dimer would have a GPI anchor. Is it even possible? Can the molecule remain anchored? Would it associate with coated pits? How would it affect membrane fluidity in such a large oligomeric complex? As presented, it is unclear if the octameric structure is a feature of the TNAP crystals but not a true reflection of the physiological structure of TNAP in the cell or in vivo.

We thank the reviewer for the question. Regarding the octameric structure of TNAP, we agree with the reviewer that it is an interesting and puzzling finding. In response to the question about how an octameric structure would function in vivo when TNAP is physiologically a GPI anchored molecule, we would like to clarify that the human TNAP exists in two major forms: a membrane-bound form with the GPI anchor and a soluble form upon the cleavage of the GPI anchor by phosphatidylinositol-specific phospholipase C (PIPLC).³⁻⁵ As shown in **Figure R5**, the spatial conflict between the membrane and neighboring TNAP subunits prevents the formation of an octamer with more than one GPI anchor, and thus the membrane-bound form should be dominated by dimeric TNAP. On the other hand, once released into circulation upon the PIPLC cleavage, TNAP can polymerize into its oligomeric form. Our structural implication on the dimeric-oligomeric form transition is supported by biochemical studies by Nosjean et al., where the oligomerization of TNAP was impeded by the presence of a GPI anchor and only took place in the GPI-free form.⁴ Therefore, we conclude that the octameric TNAP might primarily exist as a soluble GPI-free form, and before the cleavage of the GPI anchor, TNAP should remain in the membrane as a dimer. Although we cannot exclude the possibility of a putative membrane-bound TNAP octamer where only one subunit (A, C, E, or G) keeps its GPI moiety and thus plays the role of the membrane contact for the octamer (as illustrated in **Figure R5**), we believe that the effect of TNAP

octamer on membrane fluidity would be very limited considering that the octameric TNAP in general exists as a soluble form.

Furthermore, we appreciate the reviewer's concern about the physiological significance of TNAP octamer. We acknowledge that many polymeric structures published could represent a crystallizing effect, instead of their natural states, and thus the octamer should be carefully studied. However, it is worth noting that the high-order oligomerization of alkaline phosphatase has previously been observed since the 1980s,⁶⁻⁸ and the tetrameric or higher oligomeric TNAP was found in both organ-purified samples and recombinantly expressed TNAP.^{4,8,9} These non-crystallized forms of TNAP suggest that the octameric structure we observed in our crystal structure could potentially represent one of the natural forms of TNAP. We agree that further studies are necessary to determine the physiological relevance of the TNAP octamer, and we plan to investigate this in future research. We appreciate the reviewer's valuable feedback.

Figure R5: The octameric TNAP and GPI-anchors: **Left:** the surface model of TNAP octamer showing its GPI-anchoring sites. The eight TNAP subunits were shown as surface model and colored in yellow (protomers A, D, E, and G) and gray (protomers B, D, F, and H). The C-terminal serine residues for linkage to the GPI moieties were shown as stick model and colored in red (protomers A and G) and green (protomers B and H). **Right:** a proposed working model for TNAP octamerization. This procedure started from the dimeric membrane-bound form with both protomers bearing GPI. The enzymatic cleavage will remove the GPI anchor and release the dimeric TNAP into the extracellular environment, thereby forming a soluble octamer. Alternatively, a dimeric TNAP with only one GPI anchor could recruit soluble dimeric TNAP to form a putative membrane-bound octamer.

A similar concern relates to the development of an antibody that apparently enhances catalytic activity. A careful characterization of why/how antibody binding might be affecting the catalytic activity is lacking. From the depiction of the Sc-Fv binding shown in Figure 4 it is apparent that antibody binding would be difficult onto a GPI-anchored TNAP since the antibody seems to attach very close to the C-terminus where the GPI anchor would be covalently attached. Have the authors tested the binding of this antibody to native membrane-bound TNAP? Does it enhance activity under those

circumstances?

Thanks for the interesting question. Regarding the first concern, we acknowledge that we do not have a clear understanding of the mechanism underlying the antibodies' ability to enhance TNAP activities. The epitope of the antibody is far away from the calcium-binding site and active site, excluding the possibility of a direct effect on the substrate entrance or the catalytic reactions. Therefore, we speculated in the manuscript that the antibody could occupy the surface S of TNAP to stabilize the structure of TNAP, and we agree that this hypothesis requires further investigation to reveal a detailed mechanism.

Regarding the second concern, we appreciate the reviewer's comment on the binding ability of our antibody to membrane-bound TNAP. As we have shown in the reply to the previous comment, TNAP could exist in membrane-bound form and soluble form, and GPI-free TNAP in the soluble form should be able to bind with our antibody. However, it will be interesting to test if our antibody can bind with TNAP in its GPI-anchored membrane-bound form. Therefore, we performed the fluorescence-activated cell sorting (FACS) analysis to determine the antibody binding to the human TNAP proteins overexpressed on the cell surface (**Figure R6**). We are delighted to report that our FACS data showed the antibody can specifically recognize the human TNAP on the cell surface, and it cannot bind with cells without TNAP.

Figure R6: The binding of JTALP001 antibody to surface-expressed TNAP. HEK293T cells (a) was used to set the gates. HEK293T cells transfected with EGFP (b) and HEK293T cells transiently transfected with the plasmids encoding the full-length hTNAP in fusion with P2A-EGFP (c) were incubated with purified JTALP001 antibody and stained with Alexa 633 labeled anti-human secondary antibody, then analyzed by FACS.

Furthermore, we have performed the cellular TNAP catalytic measurement with or without the antibody. The TNAP activity is significantly increased in presence of TNAP antibodies (**Figure R2**). Currently, we don't have evidence to show the specific form of TNAP recognized by the antibody. Before the detailed structural information about the C-terminal connecting GPI moiety gets determined, it is difficult to model the antibodies in complex with the hTNAP anchored to the cell surface. Nevertheless, the existence of a partially de-GPI TNAP dimer or the TNAP oligomer on the cell surface might provide docking surfaces for antibodies even when the membrane-bound TNAP

has its epitope covered by the cell membrane and its GPI anchor.

In short, the authors have some great data here, but there are missing out on an opportunity to describe in full detail a novel structure, referring to and contrasting conclusions from prior art, that is warranted in this context. Furthermore, they are branching into examining TNAP mutations and a novel antibody making a number of claims all of which require further validation. I would suggest that the authors put together a first paper on the structure, comparing it to PLAP and all prior art that used mutagenesis on TNAP and PLAP to help advance the field. Then perhaps move onto HPP mutations in the context of a second paper where they also have more biological proof that an octameric TNAP structure really exists in vivo. The antibody could be described in that second paper but only after the precise effect on the GPI-anchored TNAP activity had been assessed.

We appreciate the reviewer's thoughtful comments and suggestions. While we value this suggestion, we respectfully hold a different view on the suggestion to divide our findings into multiple papers. We believe that the current manuscript provides a comprehensive analysis of TNAP structure and its potential implications for disease. We have included comparisons to the prior art and discussed the implications of our findings for understanding HPP mutations. Our exploration of TNAP mutations and the novel antibody also represents an important contribution to the field. Moreover, we believe that the structural information presented in this paper is critical for understanding the pathology of HPP, a disease caused by mutant TNAP. As scientists and doctors continue to focus on patients with HPP mutations, we feel that the information presented in this manuscript will be valuable for understanding the disease and developing diagnostic methods as early as possible. In light of our commitment to providing a comprehensive analysis of the TNAP structure and its potential implications for disease, we hope not to divide our manuscript and instead present our findings in their entirety to encourage critical discussion among the experts in the field.

Reviewer #3 (Remarks to the Author):

I have been asked to review the manuscript with particular attention to the structural biology/biochemistry. The X-ray and cryo-EM structures are of high quality. Below are specific comments.

Line 116 and topics related to oligomeric state.

a) Molecular weight standards should be included with the SEC analysis (Fig S1D) to support the claim that TNAP is predominately dimeric in solution. As shown (without such calibration), the SEC analysis does not provide information on oligomeric state. Alternatively, SEC with in-line light scattering, UV, and refractive index measurements can be used to determine the molecular mass of the complex. I find the crosslinking (Fig S1B) a bit inconclusive because glutaraldehyde, which shows the most extensive crosslinking, can sometimes cause non-specific crosslinking between molecules that are not associated with one another.

We thank the reviewer for the suggestion on the confirmation of the dimeric form of TNAP. To further measure the oligomeric state of hTNAP in solution, we conduct static light scattering, and the result was showed in **Figure R7**. The LC analysis confirmed the major form of TNAP in SEC FPLC is a dimer. We agree that the glutaraldehyde crosslinking could be inconclusive and sometimes leads to non-specific crosslinking between molecules that are not associated with one another. We have therefore included the SLS results in the supplementary figures to provide additional confirmation of the oligomeric state of TNAP.

Figure R7: The static light scattering (SLS) of TNAP. The peak results showed the molar mass moments (g/mol) of M_n 1.226×10^5 ($\pm 0.326\%$) and M_w 1.227×10^5 ($\pm 0.327\%$).

b) The SEC profile of R450C (in comparison to WT) should be shown to substantiate the supposition that this mutation stabilizes the monomeric form (destabilizes the dimer). Incidentally, why was cystine chosen for this mutation, as it might be expected to cause issues due to disulfide formation? Including the reasoning for the choice of cysteine in the text would help the reader.

We thank the reviewer for the question about the biological significance of the R450C mutant and suggestions for further biochemical characterization. First of all, stabilizing TNAP in its monomeric form is challenging due to the high hydrophobicity of the

dimeric interface, which makes it difficult to stabilize the protein in its monomeric form. As a result, the severe mutations at the dimeric interface usually generate highly unstable AP mutants, which can be expressed at a very low level and hardly purified¹⁰⁻¹². Our purification efforts showed that most dimeric mutants ALPL were expressed at low levels, and no mono-dispersed peak could be collected in SEC for further biochemical assay and osteoblast studies. In our biochemical studies, mutations on R450 of ALPL are less deleterious and thus could be purified as a dimeric form which showed low activities in enzymatic assay and osteoblast study (**Figure R1&R8**). Therefore, we conclude that TNAP cannot be stabilized in the monomeric form, and the R450C mutant exists as an inactive dimeric protein, as evidenced by the wider elution peak from size-exclusion chromatography (**Figure R8**).

Figure R8: The thermostability of dimeric mutants and SEC profile of TNAP^{R450C} and TNAP^{T167M}. **Left:** The expression levels of TNAP with dimeric mutations. 1-6: immunoblotting results of HEK293T cell overexpressing hTNAP wild-type, G420S, R71H, R391H, R450C, and G420A. **Right:** The SEC FPLC profile for hTNAP^{R450C} (red), hTNAP^{T167M} (green), and hTNAP wild-type (blue).

Regarding the second comment, we chose cysteine for the R450C mutation because it is a pathogenic mutation that leads to severe symptoms in hypophosphatasia patients^{13,14}. Additionally, the dimerization of TNAP does not rely on the facilitation of a disulfide bond, and no cysteine residues are located near the dimeric interface. Therefore, we believe that the R450C mutation is unlikely to cause any artificial disulfide issues.

We appreciate the reviewer for bringing up these potential concerns, and we have made revisions to incorporate this information in the main text and supplementary figures of the manuscript.

c) The SEC profile for T167M should also be shown – it is important to consider whether the protein has a similar fold/oligomeric state to the WT protein. If the T167M mutant reduced protein stability (e.g. caused aggregation), for example, then its lack of ability to promote osteoblast mineralization could be due to misfolding rather than loss of catalytic activity.

We thank the reviewer for the suggestion. Our analysis reveals that, like the wild-type

TNAP, the SEC FPLC profile of TNAP^{T167M} exhibits a stable dimeric form of TNAP with a mono-dispersed elution peak (**Figure R8**). This result suggests that TNAP^{T167M} has an overall folding and stability comparable to that of the wild type, indicating that its inability to promote osteoblast mineralization is due to the loss of enzymatic activity rather than a structural defect.

We have incorporated the SEC FPLC profile into our manuscript and believe that it has greatly improved the quality of our work.

Metal binding sites, lines 130 and following. Justification should be given for assigning the metals to Ca/Mg/Zn ions. Were the identities of the metals determined by comparison with previous studies or by anomalous scattering etc.? If by comparison, this could be clarified in the text (with references). If by anomalous scattering, data should be shown. Omit maps are typically not sufficient to distinguish between these ions (because they have similar scattering power).

The assignment of metal ions in the TNAP structure was based on a combination of omit maps and previously reported structures of other alkaline phosphatases (APs), including *E. coli* alkaline phosphatase (ECAP) and human placental alkaline phosphatase (PLAP). In ECAP, two zinc ions and one magnesium ion were identified at the active site (PDB IDs 1ALK and 1B8J)^{15,16}, while the calcium-binding site was absent in bacteria. In PLAP, which is more similar to TNAP than ECAP, four metal ions were identified, with three ions at the active site assigned as two Zn²⁺ and one Mg²⁺. The fourth metal ion was located in a pocket at the distal side of dimeric AP and could be either Mg²⁺ or Ca²⁺ (PDB ID 1EW2)¹, but further characterization using x-ray fluorescence suggested that calcium is likely the fourth ion in PLAP² (PDB IDs 1ZEB and 1ZED). Those facts, combining the calcium dependence of TNAP activity, supported our assignment of metal ions in our structures.

We acknowledge that a more detailed discussion of the rationale for assigning the metal ions in the TNAP structure is necessary, and have incorporated this information into the manuscript. We thank the reviewer for the constructive feedback.

From the structures, the R450-D404 interaction is not between the side chains, as one would suppose from the text (line 141: “The hTNAP dimer is maintained by the static electric interactions among R71-D458, K99-D90, and R450-D404”). D404 makes a hydrogen bond with the backbone of R450. R450 is exposed to solvent; the side chain of R450 does not interact with D404. Because of the surface exposure of R450, it is unclear (from the structure alone) why the R450C mutation would disrupt dimerization. We thank the reviewer for pointing out the inaccurate description of the R450-D404 interaction. Although this interaction is mediated by a pair of oppositely charged residues, the spatial arrangement supported the reviewer’s comments, and there is a hydrogen bond, instead of electrostatic interaction, between R450 and D404. We have corrected the description at line 141 accordingly to reflect this fact.

We also appreciate the reviewer's comment on the mechanism by which the R450C mutation disrupts the function of hTNAP. Our previous responses may have shown that R450C would not severely disrupt the dimerization of hTNAP, and only a relatively

lower thermostability was suggested by the SEC profile. We acknowledge that the precise mechanism is not yet fully understood. Our crystallographic data alone cannot fully explain the effect of the R450C mutation, and additional experiments will be required to elucidate the underlying mechanism. We have revised the manuscript to better convey this uncertainty.

From inspection of the structure, R71-D458 forms an ion pair interaction between the side chains, as indicated by the authors. However, K99-D90 does not form an ion pair. Rather, K99 forms a hydrogen bond with the backbone carbonyl oxygen of T85. K99 does not interact with D90. D90 does not form a hydrogen bond with the neighboring subunit. D90 does, however, form an ion pair with R71 from the same subunit – this interaction may stabilize the R71-D458 pair between subunits.

We thank the reviewer for the comments on the interaction analysis. The distance between the ϵ -amino group of K99 and the γ -carboxyl group of D90 is about 4.1 Å, while that between the ϵ -amino group of K99 and the backbone carbonyl oxygen of T85 is about 2.8 Å. Considering the length of 4.1 Å is too long for electrostatic interaction, we revised our description as a hydrogen bond between K99-T85.

Line 144-149, the “critical role of dimerization in TNAP function” seems to be too strong of a statement. It has not been shown in this study that the indicated mutations disrupt dimerization. The authors suppose that they disrupt dimerization from their locations, but there could be other effects (e.g. expression, folding, reduced catalytic activity, etc.).

We thank the reviewer for the comments. Due to the low stability of monomeric TNAP, it is challenging to differentiate the effects of dimer disruption from other factors such as de-folding and expression. While we hypothesized that the mutations disrupt dimerization based on their locations, it is possible that other effects could be at play. However, previous pathological reports have suggested that dimeric interface mutations can have a dominant negative effect, indicating the potential importance of dimerization in TNAP function. Nevertheless, we acknowledge the uncertainty surrounding the role of the indicated mutations and agree that further functional studies are required to draw definitive conclusions. Therefore, we agree that the conclusion may be too strong, and we have revised the paragraph to reflect this uncertainty and avoid overemphasizing the role of dimerization in TNAP function.

Water molecules are not included in the atomic coordinates. The resolutions of the X-ray structures and strong (> 3 sigma) Fo-Fc densities in them would justify their addition to the X-ray structures.

Usually, Phenix software will not add water at a resolution lower than 2.8 Å. In response to the comment, we manually modeled water into the acidic TNAP structure, as the resolution was sufficient to justify their inclusion. The revised water-included structure has been provided with this re-submission for your review and examination. However, due to the limited resolution of the basic TNAP structure, we refrained from adding water molecules as it could have affected the reliability of the water molecule

assignment.

We thank the reviewer for the suggestion, which improves the quality of the structure significantly.

Supplementary Table 1. For Human TNAP18-500, basic pH (PDB ID 7YIV), the RMSD from ideality for bond lengths is listed as 0.54 Å. Perhaps this is a typo.

Average B-factors for the models could be listed in Supplementary Table 1 for protein/ligand.

We thank the reviewer for pointing out the typos and the 7YIV's RMSD from ideality for bond lengths and angles should be 0.011 Å and 1.33°, respectively. We also update the table with the average B factors for the models.

- 1 Le Du, M. H., Stigbrand, T., Taussig, M. J., Menez, A. & Stura, E. A. Crystal structure of alkaline phosphatase from human placenta at 1.8 Å resolution. Implication for a substrate specificity. *J Biol Chem* **276**, 9158-9165, doi:10.1074/jbc.M009250200 (2001).
- 2 Mornet, E. *et al.* Structural evidence for a functional role of human tissue nonspecific alkaline phosphatase in bone mineralization. *J Biol Chem* **276**, 31171-31178, doi:10.1074/jbc.M102788200 (2001).
- 3 Ciancaglini, P., Simao, A. M., Camolezi, F. L., Millan, J. L. & Pizauro, J. M. Contribution of matrix vesicles and alkaline phosphatase to ectopic bone formation. *Braz J Med Biol Res* **39**, 603-610, doi:10.1590/s0100-879x2006000500006 (2006).
- 4 Nosjean, O., Koyama, I., Goseki, M., Roux, B. & Komoda, T. Human tissue non-specific alkaline phosphatases: sugar-moiety-induced enzymic and antigenic modulations and genetic aspects. *Biochem J* **321** (Pt 2), 297-303, doi:10.1042/bj3210297 (1997).
- 5 Say, J. C., Ciuffi, K., Furriel, R. P., Ciancaglini, P. & Leone, F. A. Alkaline phosphatase from rat osseous plates: purification and biochemical characterization of a soluble form. *Biochim Biophys Acta* **1074**, 256-262, doi:10.1016/0304-4165(91)90161-9 (1991).
- 6 Unakami, S. *et al.* Molecular nature of three liver alkaline phosphatases detected by drug administration in vivo: differences between soluble and membranous enzymes. *Comp Biochem Physiol B* **88**, 111-118, doi:10.1016/0305-0491(87)90088-5 (1987).
- 7 Hawrylak, K. & Stinson, R. A. The solubilization of tetrameric alkaline phosphatase from human liver and its conversion into various forms by phosphatidylinositol phospholipase C or proteolysis. *J Biol Chem* **263**, 14368-14373 (1988).
- 8 Bublitz, R. *et al.* Heterogeneity of glycosylphosphatidylinositol-anchored alkaline phosphatase of calf intestine. *Eur J Biochem* **217**, 199-207, doi:10.1111/j.1432-1033.1993.tb18234.x (1993).
- 9 Millan, J. L. *et al.* Enzyme replacement therapy for murine hypophosphatasia. *J Bone Miner Res* **23**, 777-787, doi:10.1359/jbmr.071213 (2008).
- 10 Sone, M., Kishigami, S., Yoshihisa, T. & Ito, K. Roles of disulfide bonds in bacterial alkaline phosphatase. *J Biol Chem* **272**, 6174-6178, doi:10.1074/jbc.272.10.6174 (1997).
- 11 Numa, N. *et al.* Molecular basis of perinatal hypophosphatasia with tissue-nonspecific alkaline phosphatase bearing a conservative replacement of valine by alanine at position 406. Structural importance of the crown domain. *FEBS J* **275**, 2727-2737, doi:10.1111/j.1742-4658.2008.06414.x (2008).

- 12 Makita, S. *et al.* A dimerization defect caused by a glycine substitution at position 420 by serine in tissue-nonspecific alkaline phosphatase associated with perinatal hypophosphatasia. *FEBS J* **279**, 4327-4337, doi:10.1111/febs.12022 (2012).
- 13 Mornet, E. *et al.* Identification of fifteen novel mutations in the tissue-nonspecific alkaline phosphatase (TNSALP) gene in European patients with severe hypophosphatasia. *Eur J Hum Genet* **6**, 308-314, doi:10.1038/sj.ejhg.5200190 (1998).
- 14 Zurutuza, L. *et al.* Correlations of genotype and phenotype in hypophosphatasia. *Hum Mol Genet* **8**, 1039-1046, doi:10.1093/hmg/8.6.1039 (1999).
- 15 Kim, E. E. & Wyckoff, H. W. Reaction mechanism of alkaline phosphatase based on crystal structures. Two-metal ion catalysis. *J Mol Biol* **218**, 449-464, doi:10.1016/0022-2836(91)90724-k (1991).
- 16 Holtz, K. M., Stec, B. & Kantrowitz, E. R. A model of the transition state in the alkaline phosphatase reaction. *J Biol Chem* **274**, 8351-8354, doi:10.1074/jbc.274.13.8351 (1999).

REVIEWERS' COMMENTS

Reviewer #1 (Remarks to the Author):

This revised manuscript provides the first crystal structure of hTNAP enzyme, known when mutationally inactivated to be the cause of hypophosphatasia. This, in itself is an important contribution to the field, particularly given the comparison with other alkaline phosphatases present in this revision. In addition, the authors present evidence that a newly developed TNAP antibody can enhance TNAP activity, including a subset of hypophosphatasia mutant forms of TNAP. Finally, the authors describe an octameric structure of TNAP by crystal structure that was previously observed in tissues but not previously well defined. They state that this octameric structure is "with apparent biologic function".

The authors responded strongly to my prior suggestions, which is appreciated. The information that they present is important and potentially impactful to the field.

My only remaining comment is that, as written, the manuscript is still confusing regarding function of the octameric structure of TNAP. They state that this form is stabilizing and therefore increases TNAP expression. They state that this form does not include the GPI anchor that is important for the highly active dimeric form of membrane bound TNAP. They also state that it acts as a multifunctional unit, which would again imply high activity. Their data demonstrates that mutations in the octameric interface decrease AP activity compared to WT TNAP, although not to the extent of other hypophosphatasia associated mutations. It would be helpful if the authors could reconcile their findings on octameric TNAP, hypothesize more specifically as to its function (perhaps this form accounts for circulating TNAP?) and do so in one discussion paragraph that does not need to be lengthy. It would also be of benefit to state something regarding their findings on this octameric form in the abstract (even stating that it is an active form of TNAP would be helpful), so the reader has some understanding at the outset, of the potential significance of this octameric TNAP form.

Overall, I am highly supportive of publication of this manuscript. Though I will again leave critique of the structural studies to reviewers with expertise in that.

Reviewer #2 (Remarks to the Author):

The authors have addressed the issues raised in my review.

Reviewer #3 (Remarks to the Author):

Most of my comments have been adequately addressed. I have some minor comments regarding the revised manuscript:

Line 117: "Further analysis using size-exclusion chromatography, chemical cross-linking assay, and native gel electrophoresis indicated the hTNAP could form oligomers with various degrees of polymerization, where dimeric hTNAP dominated in solution as estimated by the static light scattering analysis, with a significant proportion forming higher oligomers (Supplementary Fig. 1b, 1d & 12)."

Crosslinking data is no longer included in the manuscript. Native gel electrophoresis can be misleading. If the authors are suggesting that their SEC profiles indicate "various degrees of polymerization", then they should indicate how the SEC profiles suggest this (with arrows etc.). There is a small leading peak at 11ml on Supp. Fig 1D. Is this what the authors are referring to? If so, it needs to be demonstrated that this peak contains hTNAP (perhaps it is already on the gel in Fig S1a? – one cannot tell from the labels of the SDS-PAGE samples). Can any molecular weight information be gained from the static light scattering analysis of what may be a similar peak (at 17 min on that SEC profile)? (If the signal is sufficient to make such analysis.)

Supplementary Figure 12. The static light scattering (SLS) of TNAP. The legend is not sufficient to describe the data (what type of column, elution volumes (or flow rate if times are shown), buffer?). Methods for SLS are not given in the methods. Have molecular weight standards been included to calibrate the SLS measurements? It would help the average reader to explain in the legend why the mass indicates a dimer (e.g. what is the expected mass of the monomer?). How is the error (which seems low) deduced?

Supplementary Figure 13. "The thermostability of dimeric mutants..." This figure does not seem to show thermostability data. (It shows expression levels and SEC profiles of selected mutants.) The authors should clarify what is meant by the phrase 'dimeric mutations'. Perhaps mutations within the dimer interface?

Reviewer #1 (Remarks to the Author):

This revised manuscript provides the first crystal structure of hTNAP enzyme, known when mutationally inactivated to be the cause of hypophosphatasia. This, in itself is an important contribution to the field, particularly given the comparison with other alkaline phosphatases present in this revision. In addition, the authors present evidence that a newly developed TNAP antibody can enhance TNAP activity, including a subset of hypophosphatasia mutant forms of TNAP. Finally, the authors describe an octameric structure of TNAP by crystal structure that was previously observed in tissues but not previously well defined. They state that this octameric structure is "with apparent biologic function".

The authors responded strongly to my prior suggestions, which is appreciated. The information that they present is important and potentially impactful to the field.

My only remaining comment is that, as written, the manuscript is still confusing regarding function of the octameric structure of TNAP. They state that this form is stabilizing and therefore increases TNAP expression. They state that this form does not include the GPI anchor that is important for the highly active dimeric form of membrane bound TNAP. They also state that it acts as a multifunctional unit, which would again imply high activity. Their data demonstrates that mutations in the octameric interface decrease AP activity compared to WT TNAP, although not to the extent of other hypophosphatasia associated mutations. It would be helpful if the authors could reconcile their findings on octameric TNAP, hypothesize more specifically as to its function (perhaps this form accounts for circulating TNAP?) and do so in one discussion paragraph that does not need to be lengthy. It would also be of benefit to state something regarding their findings on this octameric form in the abstract (even stating that it is an active form of TNAP would be helpful), so the reader has some understanding at the outset, of the potential significance of this octameric TNAP form.

Overall, I am highly supportive of publication of this manuscript. Though I will again leave critique of the structural studies to reviewers with expertise in that.

Our response: We sincerely appreciate the reviewer's supportive and insightful comments on our manuscript. We agree with the suggestion to reconcile our findings on octameric TNAP and provide a more specific hypothesis regarding its function, particularly considering its potential role in circulating TNAP. Based on the comments, we have revised the discussion and abstract accordingly to propose explanation of the function of octameric TNAP. Once again, we sincerely thank the reviewer the valuable feedback.

Reviewer #2 (Remarks to the Author):

The authors have addressed the issues raised in my review.

Our response: We sincerely appreciate the reviewer's valuable feedback and constructive comments on our manuscript.

Reviewer #3 (Remarks to the Author):

Most of my comments have been adequately addressed. I have some minor comments regarding the revised manuscript:

Line 117: "Further analysis using size-exclusion chromatography, chemical cross-linking assay, and native gel electrophoresis indicated the hTNAP could form oligomers with various degrees of polymerization, where dimeric hTNAP dominated in solution as estimated by the static light scattering analysis, with a significant proportion forming higher oligomers (Supplementary Fig. 1b, 1d & 12)."

Crosslinking data is no longer included in the manuscript. Native gel electrophoresis can be misleading. If the authors are suggesting that their SEC profiles indicate "various degrees of polymerization", then they should indicate how the SEC profiles suggest this (with arrows etc.). There is a small leading peak at 11ml on Supp. Fig 1D. Is this what the authors are referring to? If so, it needs to be demonstrated that this peak contains hTNAP (perhaps it is already on the gel in Fig S1a? – one cannot tell from the labels of the SDS-PAGE samples). Can any molecular weight information be gained from the static light scattering analysis of what may be a similar peak (at 17 min on that SEC profile)? (If the signal is sufficient to make such analysis.)

Our response: We sincerely appreciate the valuable comments provided by the reviewer. The reviewer correctly identified the 11 mL peak observed in the FPLC analysis, which indeed corresponds to the higher oligomeric state of TNAP as shown in our SDS-PAGES. We modified the Supplementary Fig 1a&d to highlight the 11 mL peak and indicate the corresponding protein band on SDS-PAGE (enclosed within a black frame).

Regarding the static light scattering (SLS) analysis, as shown in Supplementary Figure 12, a similar peak was observed at a relatively earlier elution time (approximately 17 min on the SEC profile) compared to the main peak. The estimated molecular weight of this peak was found to be approximately 2.385×10^5 g/mol, roughly twice the size of the main peak (1.230×10^5 g/mol), suggesting that it might represent the tetrameric form of TNAP. However, it is important to note that the differential refractive index (dRI) signal associated with the peak at 17 min exhibited minimal changes (as indicated by the blue curve in Supplementary Figure 12). This suggests that the data obtained from the SLS analysis may not be sufficiently reliable to draw a definitive conclusion about the oligomeric state, other than the presence of the dimeric form. Therefore, we have refrained from making any conclusive statements regarding the oligomeric state of TNAP based solely on the SLS data.

Supplementary Figure 12. The static light scattering (SLS) of TNAP. The legend is not sufficient to describe the data (what type of column, elution volumes (or flow rate if times are shown), buffer?). Methods for SLS are not given in the methods. Have molecular weight standards been included to calibrate the SLS measurements? It would help the average reader to explain in the legend why the mass indicates a dimer (e.g. what is the expected mass of the monomer?). How is the error (which seems low) deduced?

Our response: We apologize for the missing information and have revised the method part to incorporate the experimental parameters of SLS. In general, the SLS analysis was conducted using a WTC-030S5 column (Wyatt Technology) with a mobile phase containing 150 mM NaCl, 20 mM

HEPES, pH 7.5 at a flow rate of 0.5 mL min⁻¹.

Regarding the MW standard and calibration on the SLS measurements, the Multi-Angle static Light Scattering (MALS) detector is calibrated with toluene. In our analysis, the SEC system is calibrated with BSA protein in same mobile phase and operation protocol. The resultant profile was shown as following:

The estimated molecular weights for peak 1 and 2 are 6.076×10^4 and 1.158×10^5 Dalton, respectively. These estimated molecular weights are in good agreement with the expected values for BSA in its monomeric and dimeric forms, which are approximately 60.5 kD and 121 kD, respectively. This agreement further strengthens the reliability of our SLS analysis.

The reported error in our measurements is derived from fitting the data using the Zimm plot equation, which is a widely accepted method for analyzing SLS data. As mentioned, the observed error is generally low and is a result of reasonable mathematical processing.

Once again, we sincerely apologize for the missing information in the initial manuscript and we appreciate the reviewer's insightful comments, which have helped us improve the clarity and accuracy of our work.

Supplementary Figure 13. "The thermostability of dimeric mutants..." This figure does not seem to show thermostability data. (It shows expression levels and SEC profiles of selected mutants.) The authors should clarify what is meant by the phrase 'dimeric mutations'. Perhaps mutations within the dimer interface?

Our response: To clarify, we have revised the figure legend for Supplementary Figure 13 to state "Supplementary Figure 13. The effects of mutations at the dimeric interface on the expression level and SEC profile of TNAP." This revised legend accurately reflects the content of the figure, which indeed presents data on the expression levels and SEC profiles of selected mutants.

Additionally, we have also taken into account the suggestion to provide an explanation of the term "dimeric mutant" when it first appears in the legend of Figure 1. We have modified the legend of Figure 1 as follows: "The dimeric mutants of hTNAP¹⁸⁻⁵⁰⁰ proteins (TNAP with mutations at the dimeric interface) were overexpressed in HEK293T and....." This modification aims to provide a clear understanding of the term "dimeric mutant" and its association with TNAP proteins bearing mutations at the dimeric interface. We sincerely thank the reviewer for bringing this matter to our

attention, as it has undoubtedly improved the clarity of our work.